# DECOUPLING CONCEPT BOTTLENECK MODEL

## ABSTRACT

Concept Bottleneck Model (CBM) is a kind of powerful interpretable neural network, which utilizes high-level concepts to explain model decisions and interact with humans. However, CBM cannot always work as expected due to the troublesome collection and commonplace insufficiency of high-level concepts in real-world scenarios. In this paper, we theoretically reveal that insufficient concept information will induce the mixture of explicit and implicit information, which further leads to the inherent dilemma of concept and label distortions in CBM. Motivated by the proposed theorem, we present Decoupling Concept Bottleneck Model (DCBM), a novel concept-based model decoupling heterogeneous information into explicit and implicit concepts, while retaining high prediction performance and interpretability. Extensive experiments expose the success in the alleviation of concept/label distortions, where DCBM achieves state-of-the-art performance in both concept and label learning tasks. Especially for situations where concepts are insufficient, DCBM significantly outperforms other models based on concept bottleneck. Moreover, to express effective human-machine interactions for DCBM, we devise two algorithms based on mutual information (MI) estimation, including forward intervention and backward rectification, which can automatically correct labels and trace back to wrong concepts. The construction of the interaction regime can be formulated as a light min-max optimization problem achieved within minutes. Multiple experiments show that such interactions can effectively promote concept/label accuracy.

## 1 INTRODUCTION

Concept Bottleneck Model (CBM) (Koh et al., 2020; Losch et al., 2021) is an interactive and interpretable AI system, which encodes the prior expert knowledge into the neural network and makes decisions according to corresponding concepts. In detail, CBM firstly maps input images into corresponding high-level concepts, and then utilizes these concepts for downstream tasks. Moreover, CBM gives the ante-hoc explanations for the model's predictions due to its end-to-end training regimes and has been widely applied to healthcare (Chen et al., 2021; Rong et al., 2022), shift detection (Wijaya et al., 2021), algorithmic reasoning (Xuanyuan et al., 2022), and so on.

However, high-level concepts in real-world scenarios often suffer a troublesome collection and commonplace insufficiency due to the consumption of large amounts of resources. Thus, the concept information is usually not sufficient

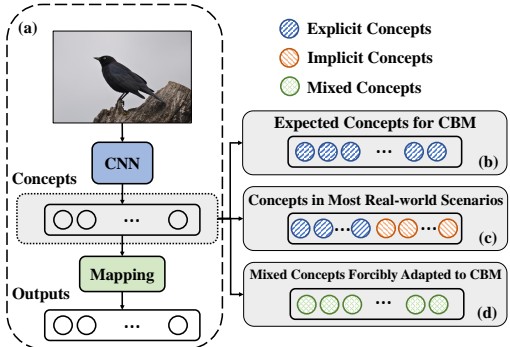

Figure 1: Concept/label distortions in CBM. (a) is the main pipeline in CBM. (b) CBM expects an ideal case with all explicit concepts. (c) Heterogeneous concepts in real-world scenarios. (d) Thus, CBM has to mix explicit/implicit concepts to achieve both concept/label learning.

to recover label information. In this circumstance, CBM cannot fit the ground-truth labels only according to the clean but insufficient concepts. On the contrary, to learn a better classifier, CBM has to inject extra information into the concept layer, i.e., sacrifice the concept accuracy (Fig. 1).

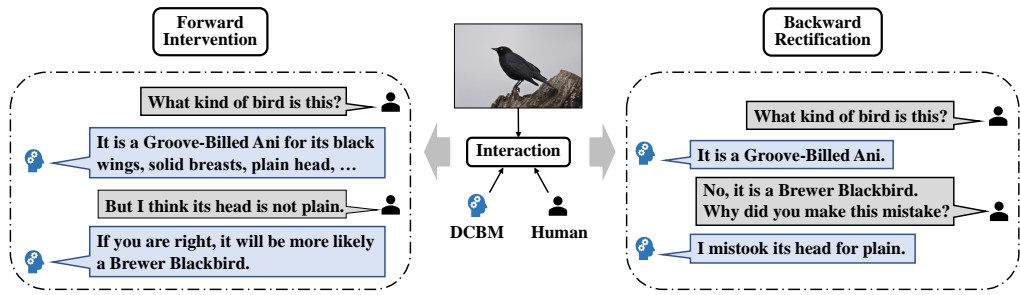

Figure 2: Two human-machine interactive tasks in DCBM. *Forward Intervention:* correcting labels according to the more accurate concepts given by human; *Backward Rectification:* tracking back to wrong concepts according to the labels updated by human.

Inspired by such observations, we theoretically reveal that CBM cannot avoid the inherent trade-off between concept and label distortions due to the mixture of explicit/implicit information.

Motivated by the theoretical results, we propose Decoupling Concept Bottleneck Model (DCBM), a novel concept-based model, to decouple the heterogeneous information into explicit and implicit concepts. In detail, DCBM automatically allocates the implicit concepts to auxiliary neurons, as such the pollution of explicit concepts can be avoided. Furthermore, DCBM maximizes the interpretability of the models via the Jensen-Shannon (JS) divergence constraint during training.

Equipped with the DCBM, we aim to conduct corresponding human-machine interactive tasks, regarded as one of the most important components for such concept-based models. However, the design of interaction algorithms for DCBM is not straightforward due to the correlation between explicit and implicit concepts. To this end, we propose a phase-two decoupling algorithm for DCBM to peel out the explicit information via mutual information (MI) estimation (Belghazi et al., 2018). We formulate it into a light min-max optimization problem solved in minutes. After the phase-two decoupling stage, we consider two interaction tasks, including forward intervention and backward rectification, which can correct labels and trace back to wrong concepts automatically via human-machine interaction (Fig. 2).

In summary, we make the following contributions:

• To our best knowledge, we are the first to theoretically reveal and prove the existence of inherent concept/label distortions in CBM for the situation where high-level concepts are insufficient.

• Motivated by the theoretical results, we propose Decoupling Concept Bottleneck Model (DCBM), a novel concept-based framework, to alleviate the distortions by automatically decoupling the concepts into explicit and implicit ones during the training process.

• Extensive experiments show that DCBM achieves state-of-the-art results in both concept and label learning tasks. In particular, compared with other concept-based models, the concept/label distortions can be alleviated conspicuously for DCBM with inadequate concepts.

• We devise a novel comprehensive human-machine system, which can automatically correct labels and ascertain wrong concepts via feedback from human experts. The system is designed to decouple heterogeneous information by solving a light MI-based optimization problem within minutes.

## 2 THEORETICAL RESULTS

We firstly introduce Concept Bottleneck Model (CBM) (Koh et al., 2020), and then present the dilemma of concept/label distortions in CBM theoretically when the concepts are insufficient.

### 2.1 CONCEPT BOTTLENECK MODEL (CBM)

Given $N$ training triples $\{\boldsymbol{x}_n, \boldsymbol{c}_n, \boldsymbol{y}_n\}_{n=1}^N$, where $\boldsymbol{x}_n \in \mathbb{R}^D, \boldsymbol{c}_n \in \mathbb{R}^d, \boldsymbol{y}_n \in \mathbb{R}^k$ respectively indicate input, concept, and label vector. With $g : \mathbb{R}^D \to \mathbb{R}^d$ mapping from input space into concept space and $f : \mathbb{R}^d \to \mathbb{R}^k$ mapping from concept space into label space, CBM defines two loss functions, i.e., $\mathcal{L}_Y$ and $\mathcal{L}_C$, which measure the difference between the ground-truth labels (concepts)

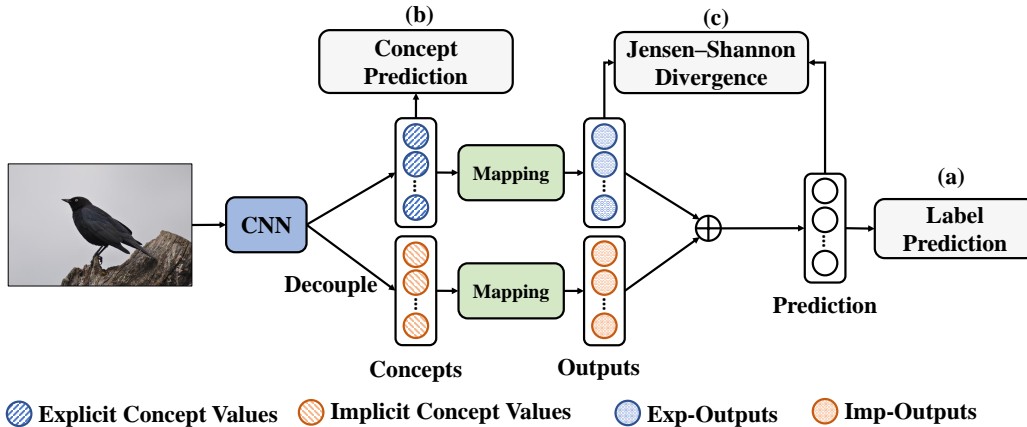

Figure 3: Pipeline of the proposed DCBM. 'Exp-Outputs' and 'Imp-Outputs' represent the outputs of explicit and implicit concepts, respectively. We jointly consider three objectives, including a) Label prediction, to optimize the label accuracy; b) Concept prediction, to optimize the concept accuracy; c) JS Divergence, to force the model to make predictions according to explicit concepts. As such, the issue of forcible concept/label distortions in CBM can be automatically alleviated.

and the predicted ones, respectively. Note that CBM utilizes the supervision of concepts during training while having no access to them at test time. Specifically, CBM devises three constructions to learn the functions $g$ and $f$, i.e., *Independent*, *Sequential*, and *Joint* models, which are introduced in Appendix A in detail.

## 2.2 MAIN THEOREM

We state a simplified version of our main theorem in this section. The full version and its corresponding proof can be seen in Appendix C.

**Theorem 1 (Informal).** *Consider a dataset $\{\boldsymbol{x}_n, \boldsymbol{c}'_n, \boldsymbol{y}_n\}_{n=1}^N$ generated from the regime $\mathcal{X} \to \mathcal{C} \to \mathcal{Y}$ with insufficient concept information. Let $d_1$ ($d_2$) denote the number of known (missing) concepts, and $\mathcal{E}_Y$ ($\mathcal{E}_C$) denote the label (concept) error. Then, the following inequation holds for Joint CBM with high probability:*

$$\mathcal{E}_Y \geq \Phi(d_1, d_2) - 2\gamma \mathcal{E}_C, \tag{1}$$

*where $\gamma \in \mathbb{R}^+$, and $\Phi(d_1, d_2) \in \mathbb{R}^+$ increases (decreases) with $d_2$ ($d_1$), respectively.*

The analysis of the main theorem consists of two steps: (1) analyzing the relationship between the number of missing concepts and the label error for *Sequential* CBM; (2) revealing the trade-off between concept and label error for *Joint* CBM via the perturbation theory of least square problem (Higham, 2002). On the one hand, Theorem 1 reveals that the weighted sum of the concept error ($\mathcal{E}_C$) and label error ($\mathcal{E}_Y$) are bounded by $\Phi$. On the other hand, $\Phi(d_1, d_2)$ increases with more missing concepts $d_2$, thus the concept and label error of CBM will also increase correspondingly. Hence, Theorem 1 reveals that the concept/label trade-off will get worse with the missing concepts.

## 3 METHODOLOGY

Based on the above theorem, we argue that CBM suffers from concept/label distortions. Thus, we embark on devising a Decoupled Concept Bottleneck Model (DCBM) to relieve such distortions.

### 3.1 DECOUPLING CONCEPT BOTTLENECK MODEL

According to the Theorem 1 and its proof, CBM has to inject implicit information into the explicit one to achieve low label error, which will obviously sacrifice the concept accuracy. To decouple the aforementioned heterogeneous information, we devise an extra mapping $\tilde{f} \circ \tilde{g}$ to store the implicit

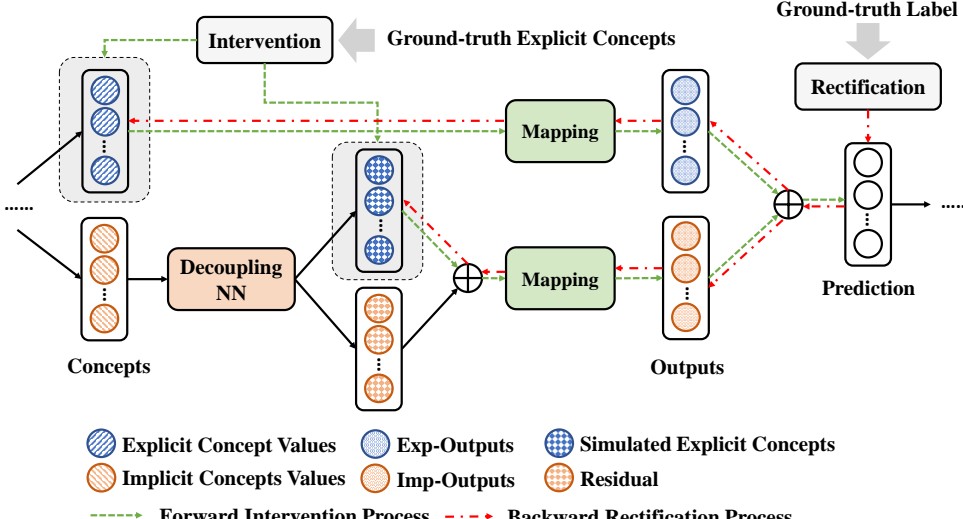

Figure 4: Forward intervention and backward rectification in DCBM. Based on MI estimation, implicit concept values are decoupled into two parts, i.e., simulated explicit concepts and residual, to extract the information of explicit concepts from implicit ones. In forward intervention, we both intervene explicit concepts and simulated explicit concepts through ground-truth explicit concepts, to promote the label prediction accuracy. In the process of backward rectification, DCBM can automatically trace back to the cause through a light optimization task.

information in the concept layer of previous CBM (Fig. 3). We consider two loss functions:

$$\mathcal{L}_1 = \sum_{n=1}^{N} \mathcal{L}_Y \big[ (f \circ g + \tilde{f} \circ \tilde{g})(\boldsymbol{x}_n), \boldsymbol{y}_n \big], \quad \mathcal{L}_2 = \sum_{n=1}^{N} \mathcal{L}_C \big[ g(\boldsymbol{x}_n), \boldsymbol{c}_n \big], \tag{2}$$

where $\tilde{g} : \mathbb{R}^D \to \mathbb{R}^{\tilde{d}}$ maps $\boldsymbol{x}_n$ into implicit concept space and $\tilde{f} : \mathbb{R}^{\tilde{d}} \to \mathbb{R}^k$ maps implicit concept values into the label space.

However, such designed loss functions cannot always protect the model's interpretability, since the predicted label might mainly depend on $\tilde{f} \circ \tilde{g}$ regardless of $f \circ g$. To ensure the model interpretability, we add an auxiliary JS Divergence constraint as follows to force the model to make decisions according to the explicit concepts as far as possible:

$$\mathcal{L}_3 = \sum_{n=1}^{N} \mathrm{JS} \big[ (f \circ g + \tilde{f} \circ \tilde{g})(\boldsymbol{x}_n) \,\|\, f \circ g(\boldsymbol{x}_n) \big]. \tag{3}$$

The overall loss is $\mathcal{L}_{\mathrm{DCBM}} = \mathcal{L}_1 + \alpha \cdot \mathcal{L}_2 + \beta \cdot \mathcal{L}_3$, where $\alpha$ and $\beta$ are hyperparameters to control the weight of different objectives. Note that the classification accuracy of explicit parts, i.e., $f \circ g$, can be guaranteed by appropriate $\beta$, while the DCBM converges to CBM when $\beta$ tends to $\infty$. However, $\beta$ is not that the higher the better, and we further analyze the choice of $\beta$ in Sec. 4.4.

**Interpretability of DCBM.** Even with extremely insufficient concept information, DCBM can guarantee the relatively low concept/label error simultaneously, however, it is impossible to expect that all model decisions can be interpreted according to the limited concepts. Under this situation, we aim to make full use of the concept information and give interpretations for as many samples as possible. To this end, we utilize the JS Divergence as a metric to evaluate the interpretability for each data point, which reflects the change of DCBM's prediction caused by implicit concepts, i.e., the black-box parts.

### 3.2 THE PROPOSED PHASE-TWO DECOUPLING ALGORITHM

In this section, we consider two human-machine interaction tasks for DCBM, including forward intervention and backward rectification (Fig. 2). The forward intervention aims to correct the label

according to the ground-truth concepts given by experts, which is proposed in (Koh et al., 2020) for the first time. Inspired by this task, we believe that a more accurate label will also help the identification of misclassified concepts, and thus, we propose backward rectification, a novel human-machine interactive task, in this paper. In detail, after DCBM makes decision $\hat{y}$ according to the corresponding concepts $\hat{c}$, the above two tasks can be formulated as follows.

**Forward Intervention.** Human experts check the learned concepts $\hat{c}$ for DCBM, and correct them to the right concepts $c^*$. Then, how would the model update its decision according to the more accurate concepts $c^*$?

**Backward Rectification.** Human experts check the learned labels $\hat{y}$ for DCBM, and tell the model that the right labels should be $y^*$. Then, how would the model trace back to wrong concepts according to the more accurate labels $y^*$ given by human experts?

Despite being equipped with clear definitions, the algorithm design is not straightforward due to the correlation between explicit and implicit concept values, and thus, we cannot ignore the explicit information hiding in the implicit one when conducting two interaction algorithms. To this end, we first separate the implicit concepts into the following two parts: (1) simulated explicit concepts, i.e., the information of explicit concepts, which can be represented as the output of a decouple function; (2) the residual, which has no or limited information of explicit concepts. The above decomposition can be obtained by an MI-based optimization problem, and we borrow the tool from MI neural estimation (Belghazi et al., 2018) to overcome the computational cost for MI estimation.

Equipped with the pre-trained decoupling neural network, it is convenient to conduct the above two interaction tasks. For forward intervention, we only need to replace the predicted concepts with expert concepts in the explicit concepts layer and the input of the decoupling network, respectively, which is in line with the green line in Fig. 4. For backward rectification, the main idea is to adjust the explicit concept values to make the model's decision agree with the expert label, which is in line with the red line in Fig. 4. The detailed description of the algorithm can be found in Appendix B.

## 4 EXPERIMENTS: CONCEPT/LABEL LEARNING

Referring to (Koh et al., 2020; Yuksekgonul et al., 2022), we perform experiments on two real-world benchmark datasets, including CUB (Wah et al., 2011) and Derm7pt (Kawahara et al., 2018), to show whether the concept/label distortions can be effectively relieved. Meanwhile, experiments on three variants of CUB are performed to show the existed concept/label distortions that CBM is suffering. We introduce the details of datasets and other baselines in Appendix D.1 and Appendix D.2.

### 4.1 CONCEPT/LABEL DISTORTIONS IN CBM

We show that there indeed exists a forcible concept/label distortions in all CBMs obviously. Take CUB dataset for example, we execute concept and label learning tasks under several cases with different ratios of concepts, namely CUB%10, CUB%20, CUB%50, as well as the original CUB with no concept discarded (Fig. 5). While $\alpha = 0.01$ is the best concept weight introduced by (Koh et al., 2020), we still present another one, i.e., $\alpha = 0.005$, performing better on label learning task. As expected, all these models undoubtedly suffer a severe distortion in either or both tasks, especially label distortion for *Independent* and *Sequential* models, and concept distortion for *Joint* model with $\alpha = 0.005$. This phenomenon inspires us to think about the underlying cause and envision an ideal model structure that can alleviate concept/label distortions.

### 4.2 METRICS AND MODEL SETTING

We introduce the metrics and model setting in Appendix D.3 and Appendix D.4, respectively.

### 4.3 BENCHMARKING MODEL ACCURACY

DCBM is compared with concept-based models, including *Independent*, *Sequential*, *Joint*, Concept-based Model Extraction (Kazhdan et al., 2020) (*CME*), Concept Bottleneck Model with Additional Unsupervised Concpets (Sawada & Nakamura, 2022) (*CBM-AUC*), and some models without interpretability, including *Standard* models with/without bottleneck, and *Multitask* model (Table 1).

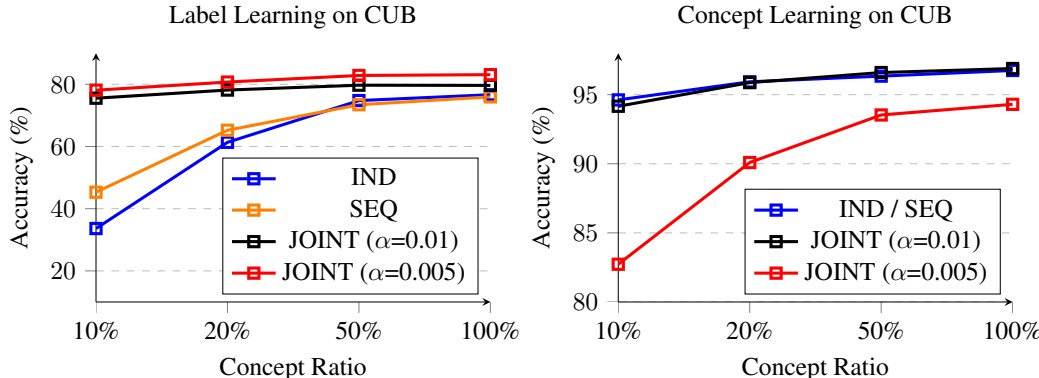

Figure 5: Label (left) / Concept (right) prediction accuracy on CUB when concepts are randomly dropped. We compare different cases with 10%/20%/50%/100% concepts. As expected, *Independent* (IND), *Sequential* (SEQ), and *Joint* models suffer an obvious distortion in either or both tasks.

Though *Joint* model had little trade-off between label and concept when concepts are sufficient, we still find the amplified concept/label distortions with more concepts invisible. To verify this, we give two groups of concept weights to ensure the best performance in either label or concept learning task, which are $\alpha = 0.01/0.005$ in *Joint* model and $\alpha = 0.1/0.01$ in DCBM. The results in Table 1 indicate that DCBM outperforms all other approaches, and even surpasses the models with no interpretability. Note that compared with other concept-based models, with more ratios of concepts discarded on CUB, the performance decline in both tasks is alleviated obviously for DCBM. Especially for DCBM with $\alpha = 0.1$, the decline of label and concept learning tasks is **-1.09%** and 0.85%, respectively, while it is 4.11%/2.73% in *Joint*-0.01, 5.00%/11.57% in *Joint*-0.005, 42.93%/2.13% in *Independent* model, 30.65%/2.13% in *Sequential* model, 40.41%/1.21% in *CME*, and 0.70%/1.74% in *CBM-AUC*. Moreover, DCBM-0.1 significantly outperforms other concept-based models in both concept and label learning tasks on datasets with insufficient concepts, i.e., CUB20%, CUB10%, Derm7pt.

The results of DCBM-0.1 in Table 1 show a counterintuitive but interesting phenomenon that label accuracy even increases with higher ratios of concepts discarded on CUB, which is contrary to any other approach. To illustrate this, we together consider label accuracy of DCBM-0.1, DCBM-0.01, and *Standard* model without bottleneck (abbreviated as STAN). It is intuitive that with more ratios of concepts discarded on CUB, both DCBM-0.1 and DCBM-0.01 will lose interpretability and converge to STAN, no wonder the label accuracy of DCBM-0.1 and DCBM-0.01 converges to the one of STAN, which is 82.54%. Therefore, the reason in our conjecture that the label accuracy of DCBM-0.1 decreases is because $\alpha = 0.1$ is so large that it forces the model to learn concepts better while, to some extent, affecting the label accuracy. However, for DCBM-0.01, $\alpha = 0.01$ is small and under this situation, the concepts can promote label accuracy.

## 4.4 WEIGHT OF JS DIVERGENCE

Since the prediction is based on both explicit and implicit outputs, it is an interesting question how the explicit and implicit outputs themselves influence the final prediction. Fig. 6 shows the Label (a-e) / Concept (f) accuracy for DCBM with $\alpha = 0.1$ on Derm7pt, CUB, and three variants of CUB, where EXP-DCBM and IMP-DCBM present the prediction results only harnessing explicit and implicit outputs, respectively. The results show that EXP-DCBM keeps a very low label accuracy when the weight of JS Divergence is close to 0 and gradually transcends with the rise of weight. By contrast, IMP-DCBM has a rather higher label accuracy than EXP-DCBM when the weight of JS Divergence is close to 0, and keeps stable or increases slightly with the rise of weight, but suddenly drops to a very low degree when the weight achieves a very high level, e.g., $\beta$=10.0 on CUB and its variants. We argue that a large weight of JS Divergence forces EXP-DCBM to be close to DCBM, and thus might damage the influence of IMP-DCBM. In Appendix E.4, we also illustrate a counterintuitive phenomenon why the lable accuracy of IMP-DCBM rises first and then falls precipitously as shown in Fig. 6 (a-c,e). However, though the label accuracy for DCBM's gradients fluctuates severely, the label accuracy for DCBM itself remains stable, with the exception

Table 1: Accuracy (%) of concept and label learning on CUB, Derm7pt, and three variants of CUB. We compare DCBM with interpretable (INTE.) baselines, including *Independent* (IND), *Sequential* (SEQ), *Joint*, *CME*, and *CBM-AUC* models. Meanwhile, a series of black-box models are compared due to their high performance, including *Standard* models with bottleneck (STAN-BNK) / without bottleneck (STAN), and *Multitask* (MULTI) model. To display better performance on either task, we employ two groups of concept weights in *Joint* model ($\alpha = 0.01/0.005$) and DCBM ($\alpha = 0.1/0.01$). The best results in each task are in **bold**. Our methods are marked brown.

| TASK | INTE. | APPROACH | DATASET | | | | |
| --- | --- | --- | --- | --- | --- | --- | --- |
| | | | **CUB** | **CUB%50** | **CUB%20** | **CUB%10** | **DERM7PT** |
| LABEL | ✓ | IND | $76.65 \pm 0.22$ | $74.76 \pm 1.61$ | $61.34 \pm 0.61$ | $33.72 \pm 2.12$ | $71.28 \pm 1.32$ |
| | | SEQ | $75.95 \pm 0.26$ | $73.43 \pm 0.76$ | $65.22 \pm 0.39$ | $45.30 \pm 5.44$ | $75.68 \pm 0.39$ |
| | | JOINT-0.01 | $79.64 \pm 0.18$ | $79.71 \pm 0.67$ | $78.15 \pm 0.30$ | $75.53 \pm 0.29$ | $83.02 \pm 0.29$ |
| | | JOINT-0.005 | $83.10 \pm 0.28$ | $82.84 \pm 0.03$ | $80.66 \pm 0.07$ | $78.10 \pm 0.36$ | $82.29 \pm 0.36$ |
| | | CME | $78.33 \pm 0.21$ | $74.53 \pm 0.04$ | $61.78 \pm 0.97$ | $37.92 \pm 0.39$ | $79.45 \pm 1.48$ |
| | | CBM-AUC | $81.26 \pm 0.37$ | $81.20 \pm 0.36$ | $81.26 \pm 0.23$ | $80.56 \pm 0.43$ | $75.47 \pm 2.83$ |
| | | (ours) DCBM-0.1 | $81.01 \pm 0.40$ | $81.34 \pm 0.20$ | $81.58 \pm 0.51$ | $82.10 \pm 0.05$ | $\mathbf{85.32 \pm 0.39}$ |
| | | (ours) DCBM-0.01 | $\mathbf{83.95 \pm 0.12}$ | $\mathbf{83.53 \pm 0.17}$ | $\mathbf{82.87 \pm 0.33}$ | $\mathbf{82.73 \pm 0.31}$ | $85.12 \pm 1.04$ |
| | ✗ | STAN-BNK | $81.93 \pm 0.20$ | $81.56 \pm 0.44$ | $80.39 \pm 0.50$ | $77.92 \pm 0.24$ | $83.23 \pm 2.43$ |
| | | STAN[#] | $82.54 \pm 0.17$ | $82.54 \pm 0.17$ | $82.54 \pm 0.17$ | $82.54 \pm 0.17$ | $82.18 \pm 1.67$ |
| | | MULTI | $83.43 \pm 0.22$ | $82.80 \pm 0.14$ | $81.35 \pm 0.08$ | $78.01 \pm 0.54$ | $84.80 \pm 0.65$ |
| CONCEPT | ✓ | IND[+] | $96.75 \pm 0.03$ | $96.34 \pm 0.16$ | $95.93 \pm 0.11$ | $94.62 \pm 0.54$ | $78.69 \pm 0.49$ |
| | | SEQ[+] | | | | | |
| | | JOINT-0.01 | $96.89 \pm 0.03$ | $96.60 \pm 0.01$ | $95.87 \pm 0.03$ | $94.16 \pm 0.07$ | $73.83 \pm 2.72$ |
| | | JOINT-0.005 | $94.30 \pm 0.26$ | $93.53 \pm 0.73$ | $91.12 \pm 0.78$ | $82.73 \pm 0.99$ | $72.54 \pm 3.45$ |
| | | CME | $95.96 \pm 0.06$ | $95.78 \pm 0.06$ | $95.54 \pm 0.07$ | $94.75 \pm 0.09$ | $79.45 \pm 1.48$ |
| | | CBM-AUC | $\mathbf{97.17 \pm 0.06}$ | $96.77 \pm 0.06$ | $96.17 \pm 0.06$ | $95.43 \pm 0.06$ | $\mathbf{81.53 \pm 0.59}$ |
| | | (ours) DCBM-0.1 | $96.92 \pm 0.04$ | $\mathbf{96.90 \pm 0.07}$ | $\mathbf{96.66 \pm 0.03}$ | $\mathbf{96.07 \pm 0.02}$ | $80.70 \pm 0.22$ |
| | | (ours) DCBM-0.01 | $96.30 \pm 0.12$ | $96.14 \pm 0.15$ | $94.89 \pm 0.35$ | $90.84 \pm 0.88$ | $76.85 \pm 1.91$ |
| | ✗ | STAN-BNK[*] | $91.34 \pm 0.15$ | $90.37 \pm 0.05$ | $87.73 \pm 0.16$ | $82.20 \pm 0.35$ | $77.06 \pm 0.47$ |
| | | MULTI | $96.10 \pm 0.46$ | $95.52 \pm 0.19$ | $94.34 \pm 0.57$ | $89.13 \pm 0.90$ | $77.07 \pm 1.42$ |

[#] *Standard* models without bottleneck share the same structure regardless of the number of concepts.

[+] *Independent* and *Sequential* model have no difference in $x \rightarrow c$ process.

[*] *Standard* model is trained with concepts unsupervised, thus we inject a linear probe to the bottleneck so it can predict the concept values.

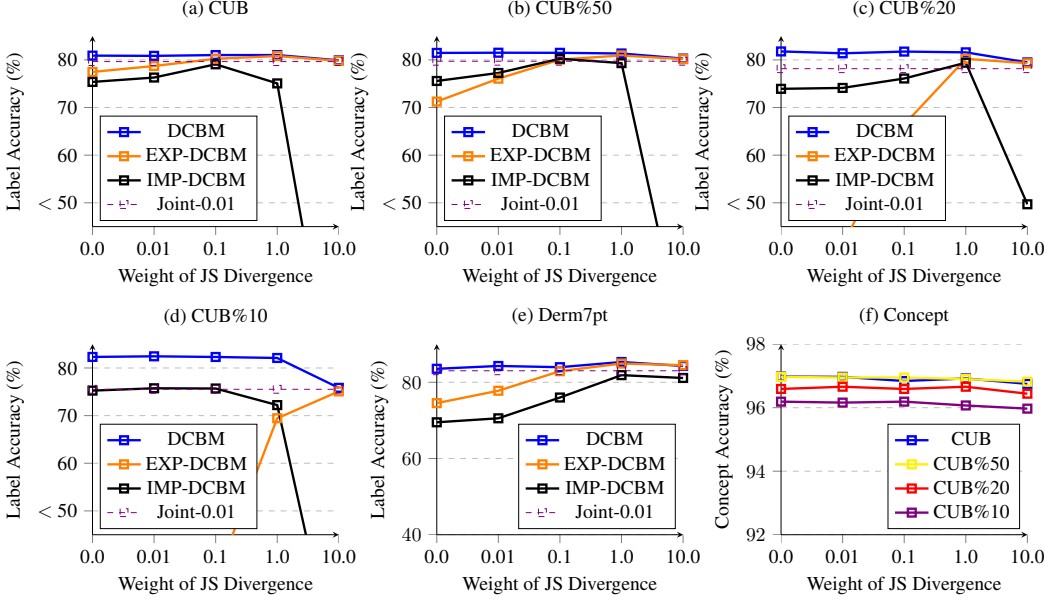

Figure 6: Label (a-e) / Concept (f) prediction for DCBM with $\alpha = 0.1$ on Derm7pt / CUB / CUB%50,%20,%10 when the weight of JS divergence varies. EXP-DCBM and IMP-DCBM represent the predictions only harnessing explicit and implicit outputs, respectively. Furthermore, we use a dotted line to express the label accuracy of *Joint-0.01*.

that the accuracy drops a little due to the nosedive of the label accuracy for IMP-DCBM. Under this perspective, we believe that the choice of $\beta$ is not arbitrary, and not always implies a trade-off

between interpretability and accuracy. For instance, the label accuracy for DCBM keeps stable or rises slightly, while the one of EXP-DCBM significantly improves when $\beta$ varies from 0.0 to 1.0. In addition to this, as shown in Fig. 6 (f), the influence of concept accuracy changes little with different weights of JS Divergence, which indicates the concept robustness in terms of $\beta$. Furthermore, the dotted lines in Fig. 6 reveal that the performance of DCBM convergences to the one of CBM when $\beta$ is sufficiently large ($\beta = 10$).

### 4.5 Other Experimental Analysis

Actually, in addition to the weight of JS Divergence, lots of other interesting analysis is worth discussing, which may either affect concept/label accuracy or interpretability and bring about some further findings. We display the relationship between the weight and values of JS Divergence in Appendix E.1 and how much the dimension of implicit concepts affects concept/label accuracy in Appendix E.2. Moreover, we compare the label accuracy predicted only by explicit concepts with the values of JS Divergence in Appendix E.3 and illustrate in Appendix E.4 why the label accuracy of IMP-DCBM rises first and then falls in Fig. 6.

## 5 Experiments: Human-machine Interaction

As introduced in Sec. 3.2, two algorithms based on MI estimation, including forward intervention and backward rectification, are devised to automatically correct label predictions and trace back to wrong concepts, while they can be achieved only by few additional training costs. In Table 2 and Table 3, we experiment the corresponding tasks for DCBM with $\alpha = 0.1$ and $\beta = 0.01/1.0$ on CUB, CUB%50, CUB%20, CUB%10, and Derm7pt. The choices of $\beta = 0.01/1.0$ respectively represent the best model performance and the best interpretability. It is also worth mentioning that we do not interact with real-world human experts in the experiments but assume that the judgments given by human experts are consistent with the concept/label values in the dataset. Therefore, we only propose a feasible plan of human-machine interaction via DCBM and it requires further examination to demonstrate whether the system can work as expected.

**Forward Intervention.** In Table 2, we display the noninterventive model performance for *Joint* model with $\alpha = 0.01$ and DCBM. Then, for CUB and its variants, we respectively intervene 11 (10% of the total 112) concept values for wrong predictions, while we intervene all the concepts related to pigment network for Derm7pt, to show the intervention promotion, see JOINT-INT and DCBM-INT. Concept values are respectively intervened to the 95th or 5th percentile of the ones over the training distribution for positive/negative samples, which is in line with (Koh et al., 2020). Besides, we perform a novel forward intervention algorithm called DCBM-INT-DEC, as introduced in Sec. 3, and as expected, the label accuracy can be further promoted in most cases. Especially for the situation where concepts are insufficient, our proposed algorithm can still provide effective intervention instead of even severely reducing the accuracy as *Joint* model does. As discussed in (Koh et al., 2020), one major reason for this phenomenon is that when the concept accuracy is low, the learned concepts are not necessarily aligned with the ground truth concepts. Similar to *CME*, the label accuracy of DCBM promotes more when more concepts are discarded. The reason is that models with poorer concept accuracy may have a larger room for label prediction improvement. Furthermore, we also conduct experiments to reveal how well the decoupling network is able to reduce the MI in Appendix E.5 and its effects during prediction in Appendix E.6.

**Backward Rectification.** Table 3 presents the unrectified model performance (DCBM) as well as the rectification without (DCBM-REC) / with (DCBM-REC-DEC) MI estimation. Results in Table 3 show the effectiveness of backward rectification, where DCBM-REC and DCBM-REC-DEC succeed in promoting concept accuracy in all cases, especially for DCBM-REC-DEC. Apart from accuracy promotion, we also give true positive rate (TPR) and false positive rate (FPR) to detail such promotion. The former represents the ability to discover wrong predictions while the latter illustrates the disturbance with correct predictions. Through in-depth observation, we find that though the TPR of DCBM-REC-DEC is low, it overpasses DCBM-DEC in prediction accuracy mainly due to a lower FPR, namely, the number of samples mistakenly corrected by DCBM-REC-DEC is far less than that of DCBM-REC. Furthermore, we need to declare that the backward rectification task is very difficult because the ratio of misclassified concepts is very low, which is less than 4% in general. Thus, our

Table 2: Forward intervention on Derm7pt, CUB and its three variants. We present the label accuracy (%) for the original *Joint* model with $\alpha = 0.01$, *CME*, and DCBM with $\alpha = 0.1$, and then respectively intervene 10% concepts to show the promotion, see JOINT-INT, CME-INT, and DCBM-INT. Moreover, we intervene simulated explicit concept values decoupled from implicit concept values (abbreviated as DCBM-INT-DEC), where label accuracy can be further improved. The best results on each dataset are in **bold.**

| APPROACH | DATASET | | | | |
|---|---|---|---|---|---|
| | CUB | CUB%50 | CUB%20 | CUB%10 | DERM7PT |
| JOINT (Koh et al., 2020) | $79.64 \pm 0.18$ | $79.71 \pm 0.67$ | $78.15 \pm 0.30$ | $75.53 \pm 0.29$ | $83.02 \pm 0.29$ |
| JOINT-INT (Koh et al., 2020) | $82.04 \pm 0.39$ | $83.06 \pm 0.10$ | $80.99 \pm 0.26$ | $67.85 \pm 0.42$ | $74.15 \pm 2.93$ |
| CME (Kazhdan et al., 2020) | $78.33 \pm 0.21$ | $74.53 \pm 0.04$ | $61.78 \pm 0.97$ | $37.92 \pm 0.39$ | $79.45 \pm 1.48$ |
| CME-INT (Kazhdan et al., 2020) | $82.43 \pm 0.21$ | $80.59 \pm 0.17$ | $71.57 \pm 0.36$ | $54.31 \pm 0.31$ | $79.56 \pm 1.60$ |
| (ours) DCBM[+] | $80.82 \pm 0.13$ | $81.49 \pm 0.32$ | $81.38 \pm 0.44$ | $82.43 \pm 0.14$ | $84.28 \pm 1.78$ |
| (ours) DCBM-INT[+] | $81.96 \pm 0.09$ | $83.39 \pm 0.77$ | $84.76 \pm 0.24$ | $86.23 \pm 0.01$ | $86.06 \pm 0.59$ |
| (ours) DCBM-INT-DEC[+] | $82.52 \pm 0.38$ | $84.32 \pm 0.71$ | $\mathbf{86.21 \pm 0.11}$ | $87.34 \pm 0.38$ | $\mathbf{86.16 \pm 0.68}$ |
| (ours) DCBM[*] | $81.01 \pm 0.40$ | $81.34 \pm 0.20$ | $81.58 \pm 0.51$ | $82.10 \pm 0.05$ | $85.32 \pm 0.39$ |
| (ours) DCBM-INT[*] | $82.59 \pm 0.25$ | $84.07 \pm 0.11$ | $85.55 \pm 0.40$ | $\mathbf{87.91 \pm 0.25}$ | $85.54 \pm 1.71$ |
| (ours) DCBM-INT-DEC[*] | $\mathbf{82.80 \pm 0.32}$ | $\mathbf{84.46 \pm 0.18}$ | $85.82 \pm 0.53$ | $87.75 \pm 0.23$ | $\mathbf{86.16 \pm 0.77}$ |

[+] DCBM with $\beta = 0.01$, typically displaying the best model performance.
[*] DCBM with $\beta = 1.0$, owning high performance while retaining the best model interpretability.

Table 3: Backward rectification on Derm7pt, CUB and its three variants. We present the original concept accuracy (ACC) of DCBM with $\alpha = 0.1$ only on samples whose label predictions are wrong. Then, we respectively show the accuracy promotion (ACC), true positive rate (TPR), and false positive rate (FPR) under an optimization regime without (DCBM-REC) / with (DCBM-REC-DEC) Decoupling Neural Network. The best results for each $\beta$ are in **bold.**

| APPROACH | METRIC | DATASET | | | | |
|---|---|---|---|---|---|---|
| | | CUB | CUB%50 | CUB%20 | CUB%10 | DERM7PT |
| DCBM[+] | | $89.36 \pm 0.05$ | $88.38 \pm 0.11$ | $87.62 \pm 0.12$ | $85.24 \pm 0.14$ | $73.37 \pm 2.16$ |
| DCBM-REC[+] | | $90.02 \pm 0.05$ | $89.73 \pm 0.12$ | $89.63 \pm 0.34$ | $85.49 \pm 0.47$ | $\mathbf{75.39 \pm 3.01}$ |
| DCBM-REC-DEC[+] | ACC (%) | $\mathbf{90.25 \pm 0.08}$ | $\mathbf{90.18 \pm 0.04}$ | $\mathbf{91.31 \pm 0.14}$ | $\mathbf{89.12 \pm 1.97}$ | $75.30 \pm 2.97$ |
| DCBM[*] | | $89.23 \pm 0.06$ | $88.48 \pm 0.22$ | $87.48 \pm 0.13$ | $85.12 \pm 0.25$ | $71.59 \pm 3.14$ |
| DCBM-REC[*] | | $90.22 \pm 0.11$ | $90.32 \pm 0.08$ | $91.22 \pm 0.42$ | $90.03 \pm 0.45$ | $72.13 \pm 3.89$ |
| DCBM-REC-DEC[*] | | $\mathbf{90.33 \pm 0.15}$ | $\mathbf{90.48 \pm 0.05}$ | $\mathbf{91.35 \pm 0.27}$ | $\mathbf{91.03 \pm 0.80}$ | $\mathbf{72.83 \pm 4.09}$ |
| DCBM-REC[+] | | $\mathbf{24.43 \pm 0.42}$ | $\mathbf{31.24 \pm 0.62}$ | $\mathbf{48.03 \pm 0.60}$ | $\mathbf{64.86 \pm 1.07}$ | $\mathbf{53.41 \pm 10.77}$ |
| DCBM-REC-DEC[+] | | $22.49 \pm 0.67$ | $27.72 \pm 0.27$ | $41.87 \pm 1.82$ | $60.44 \pm 9.90$ | $\mathbf{53.41 \pm 10.77}$ |
| DCBM-REC[*] | TPR (%) | $\mathbf{24.73 \pm 0.51}$ | $\mathbf{32.30 \pm 0.56}$ | $\mathbf{46.51 \pm 1.31}$ | $\mathbf{68.99 \pm 1.95}$ | $\mathbf{37.80 \pm 2.03}$ |
| DCBM-REC-DEC[*] | | $24.23 \pm 0.63$ | $30.99 \pm 0.88$ | $42.76 \pm 1.21$ | $66.48 \pm 3.39$ | $33.74 \pm 6.73$ |
| DCBM-REC[+] | | $2.17 \pm 0.42$ | $2.58 \pm 0.28$ | $4.49 \pm 0.41$ | $10.89 \pm 0.31$ | $22.22 \pm 2.74$ |
| DCBM-REC-DEC[+] | FPR (%) | $\mathbf{1.68 \pm 0.14}$ | $\mathbf{1.61 \pm 0.11}$ | $\mathbf{1.70 \pm 0.08}$ | $\mathbf{5.89 \pm 3.72}$ | $\mathbf{21.46 \pm 1.34}$ |
| DCBM-REC[*] | | $1.87 \pm 0.06$ | $2.12 \pm 0.16$ | $2.38 \pm 0.33$ | $6.36 \pm 0.40$ | $16.18 \pm 4.65$ |
| DCBM-REC-DEC[*] | | $\mathbf{1.69 \pm 0.07}$ | $\mathbf{1.77 \pm 0.21}$ | $\mathbf{1.70 \pm 0.12}$ | $\mathbf{4.74 \pm 1.21}$ | $\mathbf{12.55 \pm 6.87}$ |

[+] DCBM with $\beta = 0.01$, typically displaying the best model performance.
[*] DCBM with $\beta = 1.0$, owning high model performance while retaining the best model interpretability.

main intention is to choose several predicted concepts which are possibly wrong and then query them to the human experts.

## 6 DISCUSSION AND CONCLUSION

In this paper, we theoretically reveal the inherent concept/label trade-off for CBM with insufficient concept information and develop DCBM to expand the application range for CBM. Furthermore, we propose a phase-two decoupling algorithm via MI-based optimization to implement human-machine interaction tasks, including forward intervention and backward rectification. This paper also has some limitations: (1) For the theory part, we assume $k < \min\{d_1, d_2\}$, which is not consistent with datasets which have abundant concepts; (2) We do not consider the leakage phenomenon of CBMs in our methodology. Apart from addressing the limitations, we also propose other future works as follows: (1) Extend the proposed DCBM to broad scenarios, such as graph-structured data and bioinformatics databases; (2) Utilize the concepts information with causal relationship; (3) Constrain the MI between explicit/implicit concepts during the training process.

## REPRODUCIBILITY STATEMENT

The source code to reproduce every experiment will be available to the public. See Appendix C for the proof of the main theory and Appendix D for experimental details.

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

# A  RELATED WORKS

**Concepts Bottleneck Models.**  CBMs (Koh et al., 2020; Kazhdan et al., 2020; Losch et al., 2021) extract the high-level expert knowledge from data and utilize them for downstream tasks, which aim to make the AI system more interpretable and interactive. On the one hand, post-hoc CBMs (Kazhdan et al., 2020; Yuksekgonul et al., 2022; Bai et al., 2022) generate the concept explanations from the pre-trained model, which has no access to concepts during training process. However, such methods cannot guarantee that the information of concepts can be recovered from the hidden layers of the pre-trained networks (Kazhdan et al., 2021). On the other hand, ante-hoc CBMs (Koh et al., 2020) learn the given concepts during training process to obtain informative embedding. However, several works (Kazhdan et al., 2021; Yuksekgonul et al., 2022) point out that ante-hoc CBMs suffer the catastrophic label distortion with insufficient high-level concepts. In this paper, we focus on ante-hoc CBMs and look more closely at the situations with insufficient concepts. Instead of pointing out the label distortions empirically (Kawahara et al., 2018; Yuksekgonul et al., 2022), we reveal the inherent concept/label trade-off for CBM with insufficient concepts theoretically. Furthermore, we propose DCBM to alleviate this trade-off via decoupling the heterogeneous information.

Some works also target on concept learning tasks with insufficient explicit concepts and utilize implicit information to improve the label accuracy (Li et al., 2022; Sawada & Nakamura, 2022). However, these methods do not constrain the power of implicit knowledge, which makes it impossible to give a faithful interpretation. Furthermore, several works (Bahadori & Heckerman, 2020; Goyal et al., 2019; Feder et al., 2021) also consider the causal effect of CBM during the intervention phase or model the causal relationship between concepts. However, in this paper, we do not consider the causal effect and regard it as an important future work.

**Three Training Techniques for CBM.**  CBM (Koh et al., 2020) devises three constructions to learn the functions $g$ and $f$, i.e., *Independent*, *Sequential*, and *Joint* models, which are introduced as follows:

1. *Independent* CBM learns $f$ and $g$ independently: $\hat{g} = \arg\min_g \sum_{n=1}^{N} \mathcal{L}_C[g(\boldsymbol{x}_n), \boldsymbol{c}_n]$, $\hat{f} = \arg\min_f \sum_{n=1}^{N} \mathcal{L}_Y[f(\boldsymbol{c}_n), \boldsymbol{y}_n]$. Note that $f$ is trained with the ground-truth concept $\boldsymbol{c}$, while using $\hat{g}(\boldsymbol{x}_n)$ as input at test time.

2. *Sequential* CBM learns $g$ in the same way as *Independent* CMB, but uses $\hat{g}(\boldsymbol{x}_n)$ to learn $\hat{f}$, i.e., $\hat{f} = \arg\min_f \sum_{n=1}^{N} \mathcal{L}_Y[f \circ \hat{g}(\boldsymbol{x}_n), \boldsymbol{y}_n]$.

3. *Joint* CBM learns $g$ and $f$ jointly via a hyperparameter $\alpha$ to control the weight of two parts, namely $\hat{g}, \hat{f} = \arg\min_{g,f} \sum_{n=1}^{N} \{\mathcal{L}_Y[f \circ g(\boldsymbol{x}_n), \boldsymbol{y}_n] + \alpha \mathcal{L}_C[g(\boldsymbol{x}_n), \boldsymbol{c}_n]\}$.

**Leakage in CBMs.**  Some papers (Mahinpei et al., 2021; Margeloiu et al., 2021) indicate the information leakage phenomenon in CBMs, which may impact the interpretability of the learned model. Moreover, Havasi et al. (2022) address this issue in CBMs and separate the concept information via a side-channel model to improve the leakage. Unlike the leakage of concepts caused via small perturbations, our paper focus on that CBMs fail to guarantee the concept/label accuracy simultaneously with insufficient explicit information.

**Other Concept Based Models.**  Several methods also aim to generate high-level synthetic concepts automatically during the training process (Bau et al., 2017; Kim et al., 2018; Zhou et al., 2018; Yeh et al., 2020; Parekh et al., 2021; Sarkar et al., 2022). These methods do not require explicit domain knowledge and can give explainable patterns which are responsible for the model's predictions. However, such methods are not interactive due to the synthetic concepts. In this paper, we utilize expert knowledge of the concepts directly and aim to design an interactive AI system.

# B DETAILS OF PHASE-TWO DECOUPLING ALGORITHM FOR DCBM

---

**Algorithm 1:** Phase-two Decoupling Algorithm

---

**# MI-based min-max optimization for training** $f_{\text{DEC}}$;

**Input:** Explicit (implicit) concepts $\{\hat{c}_n\}_{n=1}^{N}$ ($\{\tilde{c}_n\}_{n=1}^{N}$) learned by DCBM, decoupling
network $f_{\text{DEC}}$ and auxiliary network $f_{\text{MI}}$ with initialized parameters $\theta_{\text{DEC}}$ and $\theta_{\text{MI}}$.

**for** $E_1$ *epochs* **do**

    Sample $N$ pairs from the joint distribution $\{\hat{c}_n; \tilde{c}_n - f_{\text{DEC}}(\hat{c}_n)\}_{n=1}^{N}$;

    **for** $E_2$ *epochs* **do**

        Shuffle $N$ residuals: $\{\tilde{c}_{j_n} - f_{\text{DEC}}(\hat{c}_{j_n})\}_{n=1}^{N}$;

        Calculate the lower bound: $\mathcal{L}(\theta_{\text{MI}}) =$
        $\frac{1}{N}\sum_{n=1}^{N}[f_{\text{MI}}(\hat{c}_n; \tilde{c}_n - f_{\text{DEC}}(\hat{c}_n))] - \log[\frac{1}{N}\sum_{n=1}^{N}\exp\{f_{\text{MI}}(\hat{c}_n; \tilde{c}_{j_n} - f_{\text{DEC}}(\hat{c}_{j_n}))\}]$;

        Update $\theta_{\text{MI}}$: $\theta_{\text{MI}} = \text{optim.Adam}(\theta_{\text{MI}}, -\nabla\mathcal{L}(\theta_{\text{MI}}))$;

    Shuffle $N$ residuals: $\{\tilde{c}_{k_n} - f_{\text{DEC}}(\hat{c}_{k_n})\}_{n=1}^{N}$;

    Calculate the loss for decoupling network: $\mathcal{L}(\theta_{\text{DEC}}) = \frac{1}{N}\sum_{n=1}^{N}[f_{\text{MI}}(\hat{c}_n; \tilde{c}_n -$
    $f_{\text{DEC}}(\hat{c}_n))] - \log[\frac{1}{N}\sum_{n=1}^{N}\exp\{f_{\text{MI}}(\hat{c}_n; \tilde{c}_{k_n} - f_{\text{DEC}}(\hat{c}_{k_n}))\}] + \eta\|f_{\text{DEC}}(\hat{c})\|_2^2$;

    Update $\theta_{\text{DEC}}$: $\theta_{\text{DEC}} = \text{optim.SGD}(\theta_{\text{MI}}, \nabla\mathcal{L}(\theta_{\text{DEC}}))$;

**# Forward Intervention**;

**Input:** Explicit (implicit) concepts $\hat{c}$ ($\tilde{c}$) learned by DCBM, the ground-truth concepts $c^*$ given
by expert, and the pre-trained decoupling network $f_{\text{DEC}}$.

Calculate the residual: $r = \tilde{c} - f_{\text{DEC}}(\hat{c})$;

Update the new predicted label: $y_{\text{new}} = f(c^*) + \tilde{f}(f_{\text{DEC}}(c^*) + r)$;

**# Backward Rectification**;

**Input:** Explicit (implicit) concepts $\hat{c}$ ($\tilde{c}$) learned by DCBM, the ground-truth label $y^*$ given by
experts, and the pre-trained decoupling network $f_{\text{DEC}}$.

Initialize $c_{\text{new}}$ via $\hat{c}$: $c_{\text{new}} = \hat{c}$;

**for** $M$ *epochs* **do**

    Calculate the classification loss:
    $\mathcal{L}_Y(c_{\text{new}}) = \mathcal{L}_Y\left[f(c_{\text{new}}) + \tilde{f}(\tilde{c} + f_{\text{DEC}}(c_{\text{new}}) - f_{\text{DEC}}(\hat{c})), y^*\right]$;

    Update the concept $c_{\text{new}}$: $c_{\text{new}} = \text{optim.Adam}(c_{\text{new}}, \nabla\mathcal{L}_Y(c_{\text{new}}))$;

---

In this section, we introduce the technical detail for the design of a human-machine interactive
system in DCBM. To begin with, we decouple the information of explicit concept values $\hat{c} \in \mathbb{R}^d$
from the implicit ones $\tilde{c} \in \mathbb{R}^{\tilde{d}}$. Specifically, we assume the implicit concept values $\tilde{c}$ can be divided
into the sum of the following two parts: (1) simulated explicit concepts, i.e., the information of
explicit concepts, which can be represented as the output of a decouple function $f_{\text{DEC}} : \mathbb{R}^d \to \mathbb{R}^{\tilde{d}}$,
i.e., $f_{\text{DEC}}(\hat{c}) \in \mathbb{R}^{\tilde{d}}$; (2) the residual, i.e., $\tilde{c} - f_{\text{DEC}}(\hat{c})$, which has no or limited information of
explicit concepts. To achieve the above factorization, we minimize the MI between the explicit
concepts $\hat{c}$ and the residual term $\tilde{c} - f_{\text{DEC}}(\hat{c})$, i.e., consider the following optimization problem:

$$\min_{f_{\text{DEC}}} \text{MI}(\hat{c}; \tilde{c} - f_{\text{DEC}}(\hat{c})) + \eta\|f_{\text{DEC}}(\hat{c})\|_2^2, \tag{4}$$

where $f_{\text{DEC}}$ is a light neural network and $\eta$ is a hyperparameter to control the energy of the decou-
pling parts. Note that once the optimization problem 4 is conducted, most information of explicit
concepts hidden in implicit ones will be absorbed into $f_{\text{DEC}}(\hat{c})$, and the residual will have limited
information of $\hat{c}$. However, the above optimization problem cannot be solved directly due to the
challenge of MI estimation in high dimensions. To combat this, we borrow the tool from MI neural
estimation (Belghazi et al., 2018), and exploit the lower bound of MI to convert the calculation of
MI to the following optimization problem:

$$\text{MI}(\hat{c}; \tilde{c} - f_{\text{DEC}}(\hat{c})) \geq \max_{f_{\text{MI}}} \mathbb{E}\left[f_{\text{MI}}(\hat{c}; \tilde{c} - f_{\text{DEC}}(\hat{c}))\right] - \log\mathbb{E}[\exp\{f_{\text{MI}}(\hat{c}; \tilde{c} - f_{\text{DEC}}(\hat{c}))\}], \tag{5}$$

where $f_{\text{MI}} : \mathbb{R}^{d+\tilde{d}} \to \mathbb{R}$ is an auxiliary network for the MI estimation. Thus, the optimization
problem 4 can be converted to the following min-max optimization:

$$\min_{f_{\text{DEC}}} \max_{f_{\text{MI}}} \mathbb{E}\left[f_{\text{MI}}(\hat{c}; \tilde{c} - f_{\text{DEC}}(\hat{c}))\right] - \log\mathbb{E}[\exp\{f_{\text{MI}}(\hat{c}; \tilde{c} - f_{\text{DEC}}(\hat{c}))\}] + \eta\|f_{\text{DEC}}(\hat{c})\|_2^2. \tag{6}$$

Equipped with the pre-trained decoupling neural network $f_{\text{DEC}}$, it is convenient to conduct the above two interaction tasks. For forward intervention, we only need to replace the predicted concepts $\hat{c}$ by expert concepts $c^*$ in the explicit concepts layer and the input of decoupling network $f_{\text{DEC}}$, respectively. In detail, we update the new predicted label $y_{\text{new}}$ as follows, which is in line with the green line in Fig. 4:

$$y_{\text{new}} = f(c^*) + \tilde{f}(f_{\text{DEC}}(c^*) + \tilde{c} - f_{\text{DEC}}(\hat{c})). \tag{7}$$

For backward rectification, the main idea is to adjust the explicit concept values $\hat{c}$ to make the model's decision agree with the expert label $y^*$. In detail, we fine-tune the concept values via backpropagation, i.e., solve the following light optimization problem and update the concept $c_{\text{new}}$, which is in line with the red line in Fig. 4:

$$c_{\text{new}} = \arg\min_{c_{\text{new}}} \mathcal{L}_Y \left[ f(c_{\text{new}}) + \tilde{f}(\tilde{c} + f_{\text{DEC}}(c_{\text{new}}) - f_{\text{DEC}}(\hat{c})), y^* \right]. \tag{8}$$

## C  THEORETICAL RESULTS

This section presents the inherent dilemma of concept/label distortions in CBM theoretically. The analysis of the main theorem consists of two steps: (1) analyzing the relationship between the number of missing concepts and the label error for *Sequential* CBM; (2) revealing the trade-off between concept and label accuracy for *Joint* CBM via the perturbation theory of least square problem (Higham, 2002).

### C.1  ASSUMPTIONS

To model the above phenomenon mathematically, we firstly introduce the functional families $\mathcal{F}_{d,k}$ and the non-redundant assumption in Definition 1 and 2, respectively.

**Definition 1.** *Define $\mathcal{F}_{d,k}$ as the families of linear transform from $\mathbb{R}^d$ to $\mathbb{R}^k$:*

$$\mathcal{F}_{d,k} \triangleq \{f : c \in \mathbb{R}^d \to Wc \in \mathbb{R}^k | W \in \mathbb{R}^{k \times d}, W_{i,j} \sim \mathcal{N}(0, \tfrac{1}{d})\}. \tag{9}$$

**Definition 2.** *Let $\mathcal{C}$ be a probability distribution over $\mathbb{R}^d$. We say $\mathcal{C}$ is non-redundant if and only if:*

$$\text{MI}(\mathcal{C}_{\mathbb{A}}; \mathcal{C}_{\mathbb{B}}) < \text{H}(\mathcal{C}_{\mathbb{A}}), \quad \text{for all } \mathbb{A}, \mathbb{B} \subset \{1, 2, \cdots, d\}, \mathbb{A} \cap \mathbb{B} = \varnothing, \tag{10}$$

*where $\text{MI}(\cdot; \cdot)$ and $\text{H}(\cdot)$ denote the mutual information and entropy, respectively.*

We state our main assumption for the data generating process in Assumption 1.

**Assumption 1.** *[Two-phase Generation] The data are generated from a two-phase processes $\mathcal{G}$:*

$$\mathcal{G} : \mathcal{X} \xrightarrow{g^*} \mathcal{C} \xrightarrow{f^*} \mathcal{Y}, \tag{11}$$

*where $\mathcal{X} \subset \mathbb{R}^D$ is the distribution of the input images, $\mathcal{C} \subset \mathbb{R}^d$ and $\mathcal{Y} \subset \mathbb{R}^k$ represent the corresponding distribution of concepts and labels, respectively. $g^* : \mathcal{X} \to \mathcal{C}$ is a Lipschitz continuous concept mapping, which maps the data point to its corresponding concept, and $f^*$ is a linear classifier sampled from the families of linear transform $\mathcal{F}_{d,k}$ randomly. Furthermore, we assume that the concept distribution $\mathcal{C}$ satisfies the non-redundant assumption in Definition 2.*

*To generate the dataset, we firstly sample a data generator $G$ from the corresponding family $\mathcal{G}$. Then, we sample $N$ triples $\{x_n, c_n, y_n\}_{n=1}^N$ from the data generator $G$, and only select $d_1 < d$ concepts randomly from $\mathcal{C}$, denoted as $\{x_n, c'_n, y_n\}_{n=1}^N$ to simulate the case of insufficient concepts.*

The non-redundant assumption reveals that arbitrary concept cannot be represented by the rest ones, which guarantees the uniqueness of the linear probe in the proof of the main theorem. Furthermore, we assume the concept bottleneck model comprises two modules as follows.

**Assumption 2.** *Let $g$ and $f$ be two modules in CBM, where $g : \mathcal{X} \to \mathcal{C}$ be a powerful neural network function which can represent arbitrary Lipschitz continuous mapping and $f \in \mathcal{F}_{d,k}$ is a linear probe.*

The universal approximation assumption of neural network function can be guaranteed by the theorems in (Chen & Chen, 1995; Lu & Lu, 2020).

## C.2 MAIN RESULTS

Now we are ready to state our main theorem in this section as follows.

**Theorem 2.** *Consider a dataset $\{\boldsymbol{x}_n, \boldsymbol{c}'_n, \boldsymbol{y}_n\}_{n=1}^N$ generated from $\mathcal{G}$ described in Assumption 1, and the CBM model which satisfies the Assumption 2. Let $d_1$ $(d_2)$ denote the number of known (unknown) concepts, and assume $k < \min\{d_1, d_2\}$. Let $\mathcal{E}_C$ and $\mathcal{E}_Y$ denote the $\ell_2$ error of concepts and labels for CBM, respectively. Then, for all $\epsilon \geq 0$, the following inequation holds for Joint CBM with probability at least $1 - 2\exp\left(-\epsilon^2/2\right)$:*

$$\mathcal{E}_Y \geq \frac{1}{\sqrt{d}}(\sqrt{d_2} - \sqrt{k} - \epsilon)\|\mathbb{P}_{\boldsymbol{U}_2}(\boldsymbol{C}_2)\|_F - 2\|\boldsymbol{C}_1^\dagger\|_2\|\boldsymbol{Y}\|_F\mathcal{E}_C, \tag{12}$$

*where $\mathbb{P}_{\boldsymbol{A}}(\boldsymbol{B})$ denotes the projection of $\boldsymbol{B}$ onto the subspace $\mathrm{span}(\boldsymbol{A})$, $\boldsymbol{C}_1 \in \mathbb{R}^{N \times d_1}$ $(\boldsymbol{C}_2 \in \mathbb{R}^{N \times d_2})$ denotes the matrices of known (unknown) concepts, and $\boldsymbol{U}_2$ denotes the orthogonal complement space of $\boldsymbol{C}_1$ such that $\mathrm{span}(\boldsymbol{C}_1 + \boldsymbol{C}_2) = \mathrm{span}(\boldsymbol{C}_1) \oplus \mathrm{span}(\boldsymbol{U}_2)$.*

**Proof** Let $\boldsymbol{X} \triangleq [\boldsymbol{x}_1, \boldsymbol{x}_2, \cdots, \boldsymbol{x}_N]^\top \in \mathbb{R}^{N \times D}$, $\boldsymbol{C} \triangleq [\boldsymbol{c}_1, \boldsymbol{c}_2, \cdots, \boldsymbol{c}_N]^\top \in \mathbb{R}^{N \times d}$, $\boldsymbol{Y} \triangleq [\boldsymbol{y}_1, \boldsymbol{y}_2, \cdots, \boldsymbol{y}_N]^\top \in \mathbb{R}^{N \times k}$ be the concatenation of $N$ input images, concepts, and labels, respectively. Without loss of generality, we assume that the first $d_1$ concepts are known, and divide $\boldsymbol{C}$ into two parts, the given concepts $\boldsymbol{C}_1 \in \mathbb{R}^{N \times d_1}$ and the unknown parts $\boldsymbol{C}_2 \in \mathbb{R}^{N \times d_2}$, respectively. Applying block QR factorization on $\boldsymbol{C}$, we have:

$$\boldsymbol{C} = [\boldsymbol{C}_1\|\boldsymbol{C}_2] = [\boldsymbol{U}_1\|\boldsymbol{U}_2] \begin{bmatrix} \boldsymbol{B}_{11} & \boldsymbol{B}_{12} \\ \boldsymbol{0} & \boldsymbol{B}_{22} \end{bmatrix}, \tag{13}$$

where $\boldsymbol{U}_1 \in \mathbb{R}^{N \times d_1}$ and $\boldsymbol{U}_2 \in \mathbb{R}^{N \times d_2}$ denote two orthogonal basis matrices, and the block $\boldsymbol{B}_{11} \in \mathbb{R}^{d_1 \times d_1}$, $\boldsymbol{B}_{12} \in \mathbb{R}^{d_1 \times d_2}$ and $\boldsymbol{B}_{22} \in \mathbb{R}^{d_2 \times d_2}$ denote the coefficient matrices. Similarly, we divide the linear transform matrix $\boldsymbol{W} \in \mathbb{R}^{k \times d}$ in Assumption 1 as $\boldsymbol{W} = [\boldsymbol{W}_1\|\boldsymbol{W}_2]$, where $\boldsymbol{W}_1 \in \mathbb{R}^{k \times d_1}$ and $\boldsymbol{W}_2 \in \mathbb{R}^{k \times d_2}$ represent the linear transform matrices corresponding to $\boldsymbol{C}_1$ and $\boldsymbol{C}_2$, respectively. It is worth mentioning that $\boldsymbol{W}$ is almost surely unique due to the non-redundant and the Gaussian distribution assumptions in Assumption 1.

Firstly, we analyze the relationship between the number of missing concepts and the label error for *Sequential* CBM. Due to the universal approximation ability of the neural network function $g$, we only need to consider the following optimization problem:

$$\min_{f \in \mathcal{F}_{d_1, k}} \quad \sum_{n=1}^N \|f(\boldsymbol{c}'_n) - \boldsymbol{y}_n\|_2^2. \tag{14}$$

According to the law of data generator $\mathcal{G}$, we have:

$$\boldsymbol{Y} = \boldsymbol{C}\boldsymbol{W}^\top = [\boldsymbol{U}_1\|\boldsymbol{U}_2] \begin{bmatrix} \boldsymbol{B}_{11} & \boldsymbol{B}_{12} \\ \boldsymbol{0} & \boldsymbol{B}_{22} \end{bmatrix} \begin{bmatrix} \boldsymbol{W}_1^\top \\ \boldsymbol{W}_2^\top \end{bmatrix}. \tag{15}$$

Then, $\boldsymbol{Y}' \triangleq \boldsymbol{U}_1\boldsymbol{B}_{11}\boldsymbol{W}_1^\top + \boldsymbol{U}_1\boldsymbol{B}_{12}\boldsymbol{W}_2^\top$ and $\boldsymbol{Y}^\perp \triangleq \boldsymbol{U}_2\boldsymbol{B}_{22}\boldsymbol{W}_2^\top$ denote the optimal projection and the residual of $\boldsymbol{Y}$ onto the subspace of $\mathrm{span}(\boldsymbol{C}_1)$, respectively. Thus, we can give the lower bound for the $\ell_2$ label error of *Sequential* CBM $\mathcal{L}_Y^{\mathrm{s}}$ as follows:

$$\begin{aligned}
\mathcal{L}_Y^{\mathrm{s}} &= \|\boldsymbol{Y}^\perp\|_F^2 \\
&= \|\boldsymbol{U}_2\boldsymbol{B}_{22}\boldsymbol{W}_2^\top\|_F^2 \\
&\geq \sigma_{\min}^2(\boldsymbol{W}_2)\|\mathbb{P}_{\boldsymbol{U}_2}(\boldsymbol{C}_2)\|_F^2,
\end{aligned} \tag{16}$$

where $\sigma_{\min}(\boldsymbol{W}_2)$ denotes the least singular value of $\boldsymbol{W}_2$. Due to $\boldsymbol{W}_2 \in \mathbb{R}^{d_2 \times k}$ is the sub-matrix of $\boldsymbol{W}$, which sampled from a random matrix family satisfying the Gaussian distribution. According to the random matrix theory (Davidson & Szarek, 2001; Eldar & Kutyniok, 2012), for all $\epsilon \geq 0$, with probability at least $1 - 2\exp\left(-\epsilon^2/2\right)$ we have:

$$\sigma_{\min}(\boldsymbol{W}_2) \geq \frac{1}{\sqrt{d}}(\sqrt{d_2} - \sqrt{k} - \epsilon). \tag{17}$$

Thus, with high probability, we have:

$$\mathcal{L}_Y^{\text{s}} \geq \frac{1}{d}(\sqrt{d_2} - \sqrt{k} - \epsilon)^2 \|\mathbb{P}_{\boldsymbol{U}_2}(\boldsymbol{C}_2)\|_F^2. \tag{18}$$

Next, we consider the equivalent form of the *Joint* CBM, which can be formulated as the following optimization problem:

$$\min_{\{\hat{\boldsymbol{c}}_n\}_{n=1}^N, f \in \mathcal{F}_{d_1,k}} \mathcal{L}_Y + \alpha \mathcal{L}_C,$$

$$\mathcal{L}_C = \sum_{n=1}^N \|\hat{\boldsymbol{c}}_n - \boldsymbol{c}'_n\|_2^2; \; \mathcal{L}_Y = \sum_{n=1}^N \|f(\hat{\boldsymbol{c}}_n) - \boldsymbol{y}_n\|_2^2. \tag{19}$$

The equivalence can be guaranteed due to the universal approximation property of the neural network function. Let $\{\hat{\boldsymbol{c}}_n^*\}_{n=1}^N$ and $\hat{f}^*$ be the optimizer of the optimization problem 19, and concatenate all $\{\hat{\boldsymbol{c}}_n^*\}_{n=1}^N$ as $\hat{\boldsymbol{C}}^*$. Furthermore, let $\hat{\boldsymbol{Y}}^\perp$ denote the projection residual of $\boldsymbol{Y}$ onto the subspace span($\hat{\boldsymbol{C}}^*$). Then, according to the perturbation theory of the least square problem (Higham, 2002), we have:

$$\begin{aligned}
&\|\boldsymbol{Y}^\perp - \hat{\boldsymbol{Y}}^\perp\|_F \\
\leq &\|\hat{\boldsymbol{C}}^* - \boldsymbol{C}_1\|_F \|\boldsymbol{C}_1^\dagger \boldsymbol{Y}\|_F + \|\hat{\boldsymbol{C}}^* - \boldsymbol{C}_1\|_F \|\boldsymbol{C}_1^\dagger\|_2 \|\boldsymbol{Y}^\perp - \boldsymbol{Y}\|_F \\
\leq &2\|\hat{\boldsymbol{C}}^* - \boldsymbol{C}_1\|_F \|\boldsymbol{Y}\|_F \|\boldsymbol{C}_1^\dagger\|_2.
\end{aligned} \tag{20}$$

Then, we can obtain the lower bound of label error $\mathcal{E}_Y$ for *Joint* CBM as follows:

$$\begin{aligned}
\mathcal{E}_Y = &\|\hat{\boldsymbol{Y}}^\perp\|_F \\
\geq &\|\boldsymbol{Y}^\perp\|_F - \|\boldsymbol{Y}^\perp - \hat{\boldsymbol{Y}}^\perp\|_F \\
\geq &[\mathcal{L}_Y^{\text{s}}]^{\frac{1}{2}} - 2\|\hat{\boldsymbol{C}}^* - \boldsymbol{C}\|_F \|\boldsymbol{C}_1^\dagger\|_2 \|\boldsymbol{Y}\|_F \\
\geq &\frac{1}{\sqrt{d}}(\sqrt{d_2} - \sqrt{k} - \epsilon)\|\mathbb{P}_{\boldsymbol{U}_2}(\boldsymbol{C}_2)\|_F - 2\|\hat{\boldsymbol{C}}^* - \boldsymbol{C}_1\|_F \|\boldsymbol{C}_1^\dagger\|_2 \|\boldsymbol{Y}\|_F \\
= &\frac{1}{\sqrt{d}}(\sqrt{d_2} - \sqrt{k} - \epsilon)\|\mathbb{P}_{\boldsymbol{U}_2}(\boldsymbol{C}_2)\|_F - 2\|\boldsymbol{C}_1^\dagger\|_2 \|\boldsymbol{Y}\|_F \mathcal{E}_C.
\end{aligned} \tag{21}$$

$\square$

## C.3 DISCUSSION OF THE MAIN THEORY

Note that we assume that the weight matrix $W_2 \in \mathbb{R}^{d_2 \times k}$ is generated from a Gaussian distribution in Assumption 1, which is used when we bound the least singular value of the weight matrix. Note that the assumption can be relaxed as $rank(W_2) = k$ and we make such an assumption to obtain a more accurate description. Actually, for any matrix $W_2$, when we add a row $w \in \mathbb{R}^k$ to it, where $w$ can be regarded as the weight of a new concept, we will have:

$$\begin{bmatrix} \boldsymbol{W}_2^\top & \boldsymbol{w}^\top \end{bmatrix} \begin{bmatrix} \boldsymbol{W}_2 \\ \boldsymbol{w} \end{bmatrix} = \boldsymbol{W}_2^\top \boldsymbol{W}_2 + \boldsymbol{w}^\top \boldsymbol{w} \succeq \boldsymbol{W}_2^\top \boldsymbol{W}_2. \tag{22}$$

Thus, we can obtain $\sigma_{\min}(W_2) \leq \sigma_{\min}\left(\begin{bmatrix} W_2 \\ w \end{bmatrix}\right)$, which indicates that the decrease of concept number will increase the least singular value of the weight matrix, and hence increasing the label error according to Eq. 16.

Moreover, we assume $k < \min\{d_1, d_2\}$, which is not consistent with the CUB dataset. This is one of the limitation of our theoretical result, and we regard this problem as an important future work.

Table 4: Brief summary of the datasets. Here, # means 'the number of'. Note that the splitting of train-test-validation is in line with (Koh et al., 2020; Kawahara et al., 2018).

| DATASET | #CONCEPT | #CLASS | #TRAIN | #VAL | #TEST |
|---|---|---|---|---|---|
| CUB (Wah et al., 2011) | 112 | | | | |
| CUB%50 | 56 | | | | |
| CUB%20 | 22 | 200 | 4796 | 1198 | 5794 |
| CUB%10 | 11 | | | | |
| Derm7pt (Kawahara et al., 2018) | 8 | 2 | 688 | 322 | 636 |

## D EXPERIMENTAL PROTOCOLS

### D.1 DATASET

We consider two commonly-used datasets, including bird species and dermoscopic images, as shown in Fig. 7. Both of them express high-level concepts as such one can provide interpretability analysis. A brief summary of datasets can be seen in Table 4.

**CUB** (Wah et al., 2011) Caltech-UCSD Birds-200-2011, which we call CUB in the following text, is an extension of CUB-200 (Welinder et al., 2010), collecting a total of 200 bird species with 11,788 bird photographs. CUB comprises 312 bird attributes including wing shape, back pattern, eye color, etc. Since the concepts are noisy and sparse, we leave only 112 binary concepts instead, following the work (Koh et al., 2020), detailed in Table 5. This task aims to classify each bird photograph out of 200 different species. Thanks to the splitting standard of train/test/validation dataset (Koh et al., 2020), we only need to follow this given setting.

**Derm7pt** (Kawahara et al., 2018) Seven-Point Checklist Dermatology dataset, abbreviated as Derm7pt, is a dataset comprising 1,011 groups of images. Each group contains a clinic photograph and a dermoscopic image, respectively. Due to the lack of Seborrheic Keratosis (SK) samples, we follow the setting of the work (Lucieri et al., 2020), only considering Melanoma (MEL) and Naevi (NV) ones with a total of 823 groups. Derm7pt contains several authoritative concepts (Kittler et al., 2016) including Pigmented Networks, Streaks, etc. On account of the concept absence and interpretability on Melanoma detection, we leave 8 malignancy-associated binary concepts, including Blue Whitish Veil, Pigmented Networks, and Regression Structures, as (Yuksekgonul et al., 2022) did. We also show the concepts in Table 5. This task is a binary classification problem, judging whether an image is a melanoma or naevus. Similarly, the train/test/validation indexes are already given as well.

Note that CUB succeeds to have a large group of concepts, while Derm7pt fails. However, most datasets we have investigated typically have a small number of concepts (Nevitt et al., 2006), or have to be extracted (Tschandl et al., 2018; Rotemberg et al., 2021; Krizhevsky et al., 2009; Fong & Vedaldi, 2018) by other trained models (Lucieri et al., 2020; Abid et al., 2021). This phenomenon is understandable, since high-quality and abundant labeled concepts are labour-intensive and infrequent. This brings about a new question — What if CUB also has an insufficient group of concepts? And then, how will CBM perform under this condition? To achieve the answer, we randomly drop 50%, 80%, and 90% concepts in CUB, respectively, and form three new datasets named **CUB%50**, **CUB%20**, and **CUB%10**. We display a brief summary of the referred datasets in Table 4, while the left concepts are randomly chosen by programs and fixed as:

• **CUB50%**: 1, 2, 5, 6, 8, 10, 12, 13, 18, 19, 21, 27, 28, 32, 33, 34, 37, 39, 40, 41, 43, 46, 47, 50, 52, 54, 56, 57, 61, 62, 63, 64, 65, 66, 67, 69, 71, 72, 75, 78, 79, 80, 81, 84, 86, 92, 93, 95, 96, 98, 100, 101, 103, 104, 107, 109.

• **CUB20%**: 6, 13, 18, 28, 33, 34, 37, 39, 46, 50, 52, 54, 62, 63, 65, 66, 75, 80, 96, 98, 101, 109.

• **CUB10%**: 6, 34, 50, 52, 54, 63, 66, 98, 101, 107, 109.

### D.2 BASELINES

We adopt seven models as our baselines in this paper, including:

• *Independent* / *Sequential* / *Joint* models in CBM, as recommended in Sec. 2.1.

Table 5: Concepts used in CUB and Derm7pt.

| | Concepts in CUB | | | | | | |
|---|---|---|---|---|---|---|---|
| No. | Concept Name | No. | Concept Name | No. | Concept Name | No. | Concept Name |
| 1 | bill shape: dagger | 29 | back color: black | 57 | forehead color: yellow | 85 | back pattern: striped |
| 2 | bill shape: hooked seabird | 30 | back color: white | 58 | forehead color: black | 86 | back pattern: multi-colored |
| 3 | bill shape: all-purpose | 31 | back color: buff | 59 | forehead color: white | 87 | tail pattern: solid |
| 4 | bill shape: cone | 32 | tail shape: notched tail | 60 | under tail color: brown | 88 | tail pattern: striped |
| 5 | wing color: brown | 33 | upper tail color: brown | 61 | under tail color: grey | 89 | tail pattern: multi-colored |
| 6 | wing color: grey | 34 | upper tail color: grey | 62 | under tail color: black | 90 | belly pattern: solid |
| 7 | wing color: yellow | 35 | upper tail color: black | 63 | under tail color: white | 91 | primary color: brown |
| 8 | wing color: black | 36 | upper tail color: white | 64 | under tail color: buff | 92 | primary color: grey |
| 9 | wing color: white | 37 | upper tail color: buff | 65 | nape color: brown | 93 | primary color: yellow |
| 10 | wing color: buff | 38 | head pattern: eyebrow | 66 | nape color: grey | 94 | primary color: black |
| 11 | upperparts color: brown | 39 | head pattern: plain | 67 | nape color: yellow | 95 | primary color: white |
| 12 | upperparts color: grey | 40 | breast color: brown | 68 | nape color: black | 96 | primary color: buff |
| 13 | upperparts color: yellow | 41 | breast color: grey | 69 | nape color: white | 97 | leg color: grey |
| 14 | upperparts color: black | 42 | breast color: yellow | 70 | nape color: buff | 98 | leg color: black |
| 15 | upperparts color: white | 43 | breast color: black | 71 | belly color: brown | 99 | leg color: buff |
| 16 | upperparts color: buff | 44 | breast color: white | 72 | belly color: grey | 100 | bill color: grey |
| 17 | underparts color: brown | 45 | breast color: buff | 73 | belly color: yellow | 101 | bill color: black |
| 18 | underparts color: grey | 46 | throat color: grey | 74 | belly color: black | 102 | bill color: buff |
| 19 | underparts color: yellow | 47 | throat color: yellow | 75 | belly color: white | 103 | crown color: blue |
| 20 | underparts color: black | 48 | throat color: black | 76 | belly color: buff | 104 | crown color: brown |
| 21 | underparts color: white | 49 | throat color: white | 77 | wing shape: rounded-wings | 105 | crown color: grey |
| 22 | underparts color: buff | 50 | throat color: buff | 78 | wing shape: pointed-wings | 106 | crown color: yellow |
| 23 | breast pattern: solid | 51 | eye color: black | 79 | size: small (5 - 9 in) | 107 | crown color: black |
| 24 | breast pattern: striped | 52 | bill length: about the same as head | 80 | size: medium (9 - 16 in) | 108 | crown color: white |
| 25 | breast pattern: multi-colored | 53 | bill length: shorter than head | 81 | size: very small (3 - 5 in) | 109 | wing pattern: solid |
| 26 | back color: brown | 54 | forehead color: blue | 82 | shape: duck-like | 110 | wing pattern: spotted |
| 27 | back color: grey | 55 | forehead color: brown | 83 | shape: perching-like | 111 | wing pattern: striped |
| 28 | back color: yellow | 56 | forehead color: grey | 84 | back pattern: solid | 112 | wing pattern: multi-colored |
| | Concepts in Derm7pt | | | | | | |
| No. | Concept Name | No. | Concept Name | No. | Concept Name | No. | Concept Name |
| 1 | blue whitish veil: present | 3 | pigment network: atypical | 5 | regression structures: absent | 7 | regression structures: combinations |
| 2 | pigment network: absent | 4 | pigment network: typical | 6 | regression structures: blue areas | 8 | regression structures: white areas |

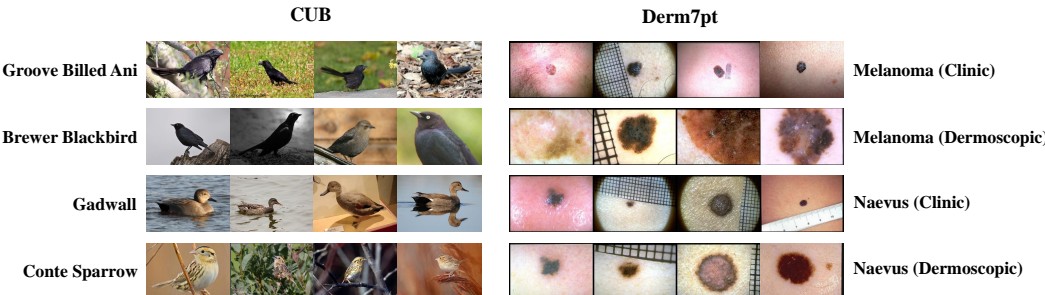

Figure 7: Examples of CUB (left) and Derm7pt (right) datasets. In CUB, each line indicates a bird species, while in Derm7pt, each line corresponds to the clinic or dermoscopic images of melanoma or nevus. The clinic and dermoscopic images are aligned up and down.

- *Standard model with / without bottleneck*, introduced in (Koh et al., 2020). The former learns $f, g$ with no concept supervised: $\hat{f}, \hat{g} = \arg\min_{f,g} \sum_{n=1}^{N} \mathcal{L}_Y[f \circ g(\boldsymbol{x}_n), \boldsymbol{y}_n]$, while the latter is devised to directly predict $x \to y$: $\hat{h} = \arg\min_h \sum_{n=1}^{N} \mathcal{L}_Y[h(\boldsymbol{x}_n), \boldsymbol{y}_n]$, with mapping $h : \mathbb{R}^D \to \mathbb{R}^k$.

- *Multitask*, also introduced in (Koh et al., 2020), is devised with an additional auxiliary loss to enable the last layer to be predictive of the concepts, which learns $g, h$ simultaneously: $\hat{g}, \hat{h} = \arg\min_{g,h} \sum_{n=1}^{N} \{\mathcal{L}_Y[h(\boldsymbol{x}_n), \boldsymbol{y}_n] + \alpha \mathcal{L}_C[g(\boldsymbol{x}_n), \boldsymbol{c}_n]\}$.

- *CME* (Kazhdan et al., 2020), also named as Concept-based Model Extraction, is based on a pre-trained deep neural network and searches a mapping from a deep layer to each concept that performs the best on concept accuracy, and then predicts the final label via traditional machine learning approaches.

- *CBM-AUC* (Sawada & Nakamura, 2022) integrates supervised concepts with unsupervised ones trained with self-explaining neural networks (Alvarez Melis & Jaakkola, 2018) in order to alleviate the restricted dimension of the concept layer.

Note that it is impossible for the *Standard* and *Multitask* model to be interpreted with the concepts ignored or uncovered by the final prediction, however, since there is no trade-off between concept and label, their performance on label prediction are competitive and worth enough attention.

Each experiment is repeated three times with mean and standard deviation reported in line with (Koh et al., 2020), and is preformed on a machine with AMD EPYC 7742 64-Core Processor CPU,

NVIDIA A100-SXM4-80GB GPU, and 1T RAM. All experiments in this paper are implemented by Pytorch (Paszke et al., 2019). **The source code will be made publicly available for reproduction requirements.**

### D.3 METRICS

Top-1 accuracy is exploited in both label and concept tasks. The former is a multi-class learning task, while the latter describes a multi-label classification task. In particular, in label prediction task, we use cross entropy loss to describe the difference between the predicted label and the ground-truth one. In concept prediction task, each concept corresponds to a binary classification mission and thus we compute the mean accuracy for all labels, where we also use a cross entropy loss to compute the difference of each predicted concept and the ground truth one during training process. This setting is in line with (Koh et al., 2020).

### D.4 MODEL SETTING

For each experiment, DCBM fine-tunes a pretrained Inception-v3 network (Szegedy et al., 2016), which maps to explicit and implicit concept values, respectively, and then connects to linear layers to obtain outputs of explicit/implicit concepts. In particular, explicit concept values are exploited to predict $x \rightarrow c$, while the sum of two outputs is used to predict the label. Apart from some general hyperparameters, e.g., learning rate, we mainly search concept weight $\alpha$ and JS Divergence weight $\beta$, to achieve high performance and maintain the interpretability. Detailed setting is shown in Appendix D.4. In short, we choose $\alpha = 0.01$ and $\alpha = 0.1$, which respectively perform better on concept and label learning tasks. Analysis about the choice of $\beta$ is further discussed in Sec. 4.4.

For the pre-processing of the training images, we adjust the size of each image in CUB and Derm7pt to $299 \times 299 \times 3$ via random jittering, horizontal flip, and cropping, which is the same setting in (Koh et al., 2020; Cui et al., 2018), while the images at test time are center-cropped and resized to 299. Similarly, but with a few differences, in DCBM training, we increase the parameter search field. We search hyperparameters on the validation dataset with $\alpha$ in [0.005, 0.01, 0.05, 0.1, 0.5, 1.0], learning rate in [0.001, 0.01], regularization strengths in [0.0004, 0.00004], reducing learning rates by 10 times after every [10, 15, 20, 30] epochs until the rates come to 0.0001. Once these parameters are fixed, we retrained models with training and validation datasets altogether until convergence with $\beta$ in [0.0, 0.01, 0.1, 1.0, 10.0]. In Table 1, we display the results of DCBM-0.1 and DCBM-0.01 using $\beta = 1.0$ on almost all datasets. The mere exception is DCBM-0.01 on CUB, where we use $\beta = 0.1$ because concepts are sufficient, and we don't need a large $\beta$ to restrict JS Divergence. Such settings can well balance the model's accuracy and interpretability, as detailed in Fig. 6. Besides, the dimension of implicit concept values is set to 256 in all cases, as we demonstrate in Appendix E.1 that the dimension of implicit concepts affects little on concept/label accuracy. The best model is chosen by either the best label accuracy or the best concept accuracy on validation dataset. Each experiment is trained with a batch size of 64 and Stochastic Gradient Descent (SGD) with a momentum of 0.9.

$f_{\mathrm{MI}}$ in Sec. 3.2 is a multilayer perception (MLP), which has neuron sizes of [*input size*, 50, 50, *output size*], where the *input size* equals to the dimension sum of explicit and implicit concept values, and *the output size* is set as 1 to compute MI. In addition, $f_{\mathrm{DEC}}$ in Sec. 3.2 is also an MLP layer with neuron sizes of [*input size*, 100, 100, 100, *output size*], where the *input size* and *output size* respectively equal to the dimensions of explicit and implicit concept values. Among each layer in $f_{\mathrm{MI}}$ and $f_{\mathrm{DEC}}$, we add activation functions $\mathrm{ReLU}(x) = \max(0, x)$ to achieve nonlinearity.

For forward intervention, to obtain a rather high performance, MI estimation optimization is trained by Adam (Kingma & Ba, 2015) with a learning rate in [0.005, 0.01, 0.05, 0.1], and the decoupling neural network is trained by SGD with a learning rate in [0.005, 0.01, 0.05, 0.1] and momentum of 0, where MI estimation is optimized for [20, 30, 40, 50] times corresponding to each decoupling process. We use the cross entropy on training datasets to perform early stopping. Besides, the backward rectification is learned by Adam with a learning rate in [0.0025, 0.025] and normalization weight in [0, 0.01, 0.1] until convergence. The best models are chosen via the best performance on validation datasets.

Table 6: Time overheads (seconds) on the training process of concept/label prediction (DCBM), decoupling neural network (DEC), forward intervention (DCBM-INT-DEC), and backward rectification (DCBM-REC-DEC) , which are recorded over an entire training process.

| APPROACH | DATASET | | | | |
|---|---|---|---|---|---|
| | **CUB** | **CUB%50** | **CUB%20** | **CUB%10** | **DERM7PT** |
| DCBM[+] | $33260 \pm 5212$ | $21920 \pm 9138$ | $26264 \pm 6824$ | $14654 \pm 4991$ | $3436 \pm 788$ |
| DEC | $362 \pm 78$ | $312 \pm 83$ | $210 \pm 35$ | $147 \pm 34$ | $491 \pm 61$ |
| DCBM-INT-DEC[*] | $< 0.01$ | $< 0.01$ | $< 0.01$ | $< 0.01$ | $< 0.01$ |
| DCBM-REC-DEC[*] | $0.27 \pm 0.01$ | $0.34 \pm 0.02$ | $0.47 \pm 0.02$ | $0.67 \pm 0.10$ | $0.39 \pm 0.13$ |

[+] High standard deviations are mainly due to the early stopping strategy during training.
[*] We show the time overhead of each intervened/rectified sample.

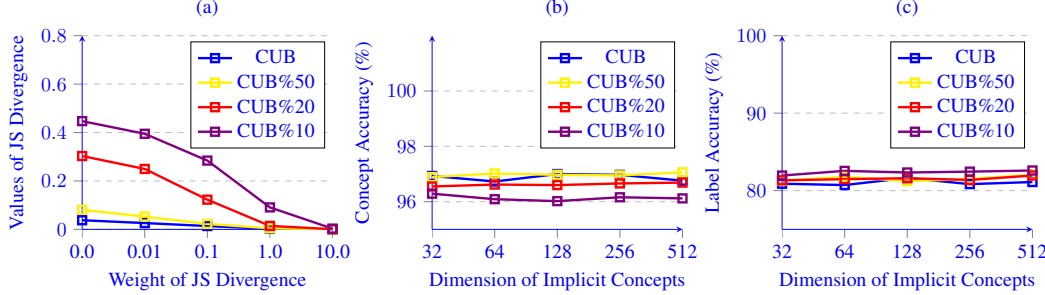

Figure 8: (a) The relationship between $\beta$ and the values of JS Divergence on CUB and its three variants. (b-c) The impact that different dimensions of implicit concepts [32, 64, 128, 256, 512] have on concept (b) / label (c) accuracy on CUB and its three variants with $\alpha = 0.1$ and $\beta = 1.0$.

## D.5 TIME OVERHEAD FOR HUMAN-MACHINE INTERACTIVE SYSTEM

The interaction system is based on a light min-max optimization introduced in Appendix B, which can be achieved within minutes. Equipped with the pretrained decoupling networks, DCBM can respectively perform forward intervention and backward rectification within 0.01s and 1s for each sample. Detailed time overheads are shown in Table 6, where we choose $\alpha = 0.1$ and $\beta = 1.0$ as a typical example in all tasks. Looking from the table it is apparent that the forward intervention and backward rectification process are much quicker than model training.

## E  EXPERIMENTAL ANALYSIS

### E.1 HOW DOES THE WEIGHT OF JS DIVERGENCE AFFECT ITS VALUES?

In the methodology section (Sec. 3), we use $\beta$ to force the prediction from explicit concepts not to be far away from the one from both explicit and implicit concepts. An interesting question is whether $\beta$ does achieve such expectations. To evaluate this, in Fig. 8 (a), we detail the relationship between the weight of JS Divergence and its values. When $\beta = 0$, fewer discarded concepts lead to higher values of JS Divergence; however, with $\beta$ rising from 0.0 to 10.0, the values of JS Divergence correspondingly decrease to 0 in all cases, which illustrates that by choosing a higher $\beta$, the prediction from explicit concepts can be effectively controlled to be almost the same as the prediction from both explicit and implicit concepts.

### E.2 HOW THE DIMENSION OF IMPLICIT CONCEPTS AFFECTS CONCEPT/LABEL ACCURACY?

Hyperparameters like the dimension of implicit concepts $\tilde{d}$ can intuitively affect the model prediction. Under this perspective, in Fig. 8 (b-c), we take $\alpha = 0.1$ and $\beta = 0.01$ as an example and depict how much the dimension of implicit concepts affects concept/label accuracy on CUB and its three variants; however, both Fig. 8 (b-c) show that the dimension of implicit concepts has little impact on concept/label accuracy, despite some slight fluctuation reasonably exists. This phenomenon shows the robustness of DCBM with respect to $\tilde{d}$.

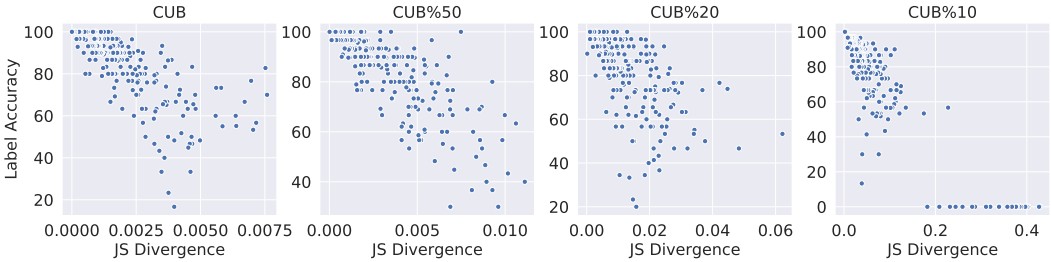

Figure 9: The relationship between JS Divergence and the label accuracy predicted on by explicit concepts on CUB and its three variants. Each point represents a concrete category.

### E.3 WHETHER JS DIVERGENCE RELATES TO INTERPRETABILITY?

It's worth discussing whether the JS Divergence can evaluate the model interpretability. To address this, in Fig. 9, we display the relationship between JS Divergence and the label accuracy predicted by explicit concepts on CUB and its three variants. Each point in the figure represents a concrete bird identification. Looking from an overall perspective it is apparent that the JS Divergence is generally larger when more concepts are discarded, which indicates that the interpretability is weakened when concepts are insufficient. This phenomenon demonstrates that the JS Divergence is an effective metric to evaluate the model interpretability. In short, DCBM cannot give explanations for all data points with insufficient concepts and it is also impossible for other models. However, DCBM can give accurate predictions for data via explicit/implicit parts and meanwhile judge whether the model decision for this data point can be interpreted by given concepts via JS Divergence. If the answer is yes, DCBM will give the corresponding interpretation.

### E.4 WHY DOES THE LABEL ACCURACY OF IMP-DCBM RISE FIRST AND THEN FALL?

It's intuitive that with the increase of the weight of JS Divergence $\beta$, explicit concepts have a greater impact on label learning, and thus, implicit ones are supposed to be less informative about labels; however, we fail to see such intuition in Fig. 6 (a-c, e) and find the label accuracy of IMP-DCBM rises first and then falls precipitously. To illustrate this counterintuitive phenomenon, we present some detailed information in Fig. 10.

Actually, by increasing $\beta$, the distribution of explicit outputs can be forced to be similar to the sum of explicit and implicit outputs, which leads to two consequences: 1) the implicit outputs become similar to the explicit ones; 2) the implicit outputs tend to 0. Under this perspective, we use the JS Divergence of explicit and implicit outputs in Fig. 10 (a) to illustrate the former and the Mean Absolute Value (MAV) of implicit outputs in Fig. 10 (b) to illustrate the latter. Looking first of all at CUB in Fig. 10 (a), the JS Divergence keeps stable from 0.0 to 0.1, so it's reasonable for the label accuracy of IMP-DCBM rises together with EXP-DCBM, but it finally drops because the JS Divergence surges. By comparison, there's little rise in the label accuracy of IMP-DCBM when $\beta$ changes from 0.0 to 0.1 on CUB%10, though an obvious decline of JS Divergence exists on CUB%10. The reason is that due to the low accuracy of EXP-DCBM, the approximation to explicit outputs cannot result in a better label accuracy for IMP-DCBM. These observations are consistent with that shown in Fig. 6. Moreover, in Fig. 10 (b), the MAV of implicit concepts also converges to 0 as expected.

### E.5 HOW MUCH CAN THE DECOUPLING NETWORK REDUCE MI?

In this section, we evaluate the decrease of MI between the explicit/implicit concepts with the help of decoupling neural network. Followed by the setting in the above experiments, we utilize neural MI estimation (Belghazi et al., 2018) to approximate the MI. In detail, we calculate original MI, i.e., $MI(\hat{c}; \tilde{c})$, and the decoupled one, i.e., $MI(\hat{c}; \tilde{c} - f_{DEC}(\hat{c}))$ on CUB and its three variants for both $\beta = 0.01$ and $\beta = 1$ (see Fig. 11 (a-b)). Except for CUB%10, we can see a significant decline in MI on the other three datasets. One possible reason is that the number of concepts is too small, and thus, the decoupling network has limited room for reducing the MI.

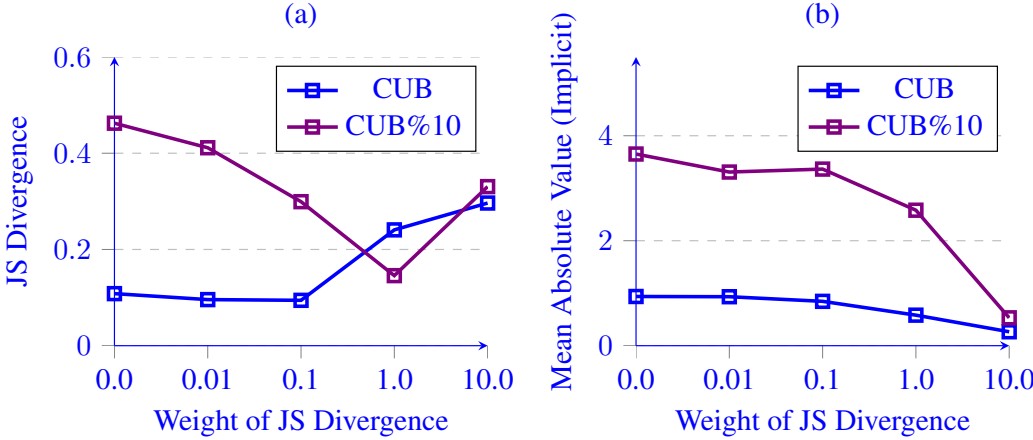

Figure 10: (a) JS Divergence of explicit and implicit outputs and (b) Mean Absolute Value (MAV) of implicit outputs on CUB / CUB%10 with $\beta$ from 0.0 to 10.0. Note that the JS Divergence in (a) is different from the one in the methodology part.

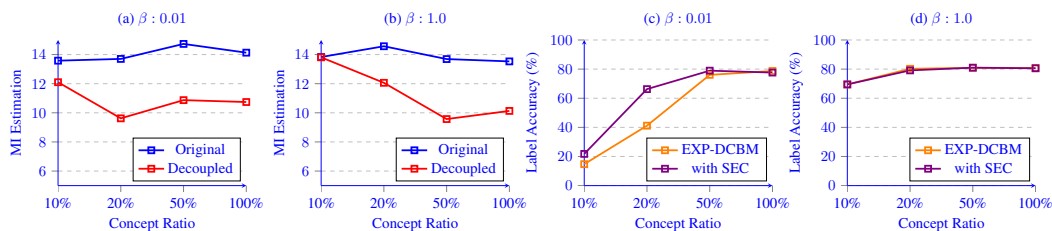

Figure 11: (a-b) MI Estimation of the original DCBM and the decoupled one using 10%/20%/50%/100% concept when (a) $\beta = 0.01$ and (b) $\beta = 1.0$. (c-d) Label accuracy of EXP-DCBM and EXP-DCBM assisted with simulated explicit concepts (abbreviated as 'with SEC') using 10%/20%/50%/100% concept when (c) $\beta = 0.01$ and (d) $\beta = 1.0$.

### E.6 WHETHER THE DECOUPLING NETWORK HELPS THE PREDICTION TASK?

Another interesting question is whether the decoupling network helps the prediction task for the explicit DCBM. Actually, DCBM can naturally combine the explicit concepts $\hat{c}$ with the simulated explicit information $f_{\mathrm{DEC}}(\hat{c})$ to make a comprehensive decision. As detailed in Fig. 11, simulated explicit information can hardly have an impact on the label prediction with a relatively higher $\beta = 1.0$ or with sufficient concepts ($\beta = 1.0$ on CUB); however, such information can effectively promote the label prediction when concepts are insufficient with a lower $\beta = 0.01$. The reason is that when $\beta$ is low and concepts are insufficient, the explicit concepts are not informative about class predictions, and thus, simulated explicit concepts have a larger room for label accuracy promotion.

