# OpenReview forum: "Decoupling Concept Bottleneck Model"
_ICLR.cc/2023/Conference — Submitted to ICLR 2023_

### Official Review · Reviewer_k8KK · 2022-10-25

**Confidence:** 3
**Correctness:** 2
**Technical Novelty And Significance:** 2
**Empirical Novelty And Significance:** 3
**Recommendation:** 6

**Clarity, Quality, Novelty And Reproducibility:**

The paper is well-written and, overall, the exposition is clear. I have already addressed my perplexities on the quantities introduced in Theorem 1 and the issues with implicit concepts in the prediction. Further clarifications are required to assess the validity of the proposed model, that in turn would improve the quality and the novelty of the manuscript. In fact, the interpretability of the model cannot be assessed directly from the experiments and it seems not to hold in general. First of all, it should be clearer from the text which parameters $\beta$ in Table 2 have been adopted for the accuracy comparison and how the interpretability of the model is assessed. I found the experiments extensive, however, the natural competitor CBM-AUC was not taken into consideration.  The details in the appendix support reproducibility, and the code repository would be available upon acceptance.


**Details Of Ethics Concerns:**




**Strength And Weaknesses:**

This paper addresses the important problem of the accuracy/interpretability trade-off of Concept-based models. In this respect, the authors prove a theorem, based on known results in Random Matrix Theory, which reveals an inherent trade-off between the concept error and the label error.  While the assumptions are a bit restrictive (the weights are assumed to be sampled independently from a gaussian distribution, which excludes real-world scenarios where correlations among weights are important) the lower bound relates the expected error on the label to that on the concepts, which is a first, relevant theoretical result for Concept-based models. Another limitation is inherited by the dimensionality of the classification space, which is assumed to be $k < \min(d_1, d_2)$, where $d_1$ and $d_2$ refer to the dimension of the explicit and implicit concepts, respectively. It seems that the CUB dataset violates the assumptions of the Theorem, being not applicable in that scenario (in fact, in the CUB-100% dataset d_1=112, where k=200).

Following, motivated by the theorem, the authors suggest that decoupling the explicit concepts and implicit concepts should alleviate the overall label/concept distortion. Upon carefully analyzing the result of the theorem, it is still not clear to me how this claim is effectively supported. In fact, the quantities in Eq. (10) are neither evaluated nor accounted for in the formulation of the model.
As it stands, the theorem yields an interesting result but it seems not so related to the model proposed by the authors nor to the experimental evaluation.

The experimental evaluation of the label/concept accuracy of the model supports the major performances of the model w.r.t. other Concept-based proposals.
For what concerns the interpretability of the model, a key requirement of CBM is the prediction to depend mostly on the interpretable (in this case explicit) concepts.
As stated by the authors, an important parameter of the DCBM is the weight $\beta$ of the Jensen-Shannon (JS) divergence, which regulates the influence of the implicit concepts for the prediction. To retain interpretability, the prediction should depend mostly on explicit concepts and, in principle, high values of $\beta$ should enforce this property. This holds as long as the problem of predicting $Y$ given $\hat C$ is linearly separable. Does this condition hold with the selected concepts in the CUB experiments? If not, the explicit concepts themselves would not suffice to predict correctly the label; hence optimizing the JS divergence would naturally deteriorate the model accuracy. When unsupervised concepts are relevant for prediction, they should be somehow interpretable, otherwise, the interpretability of the model falls off. For example, in Concept Bottleneck Model With Additional Unsupervised Concepts (in short CBM-AUC) by Sawada and Nakamura, IEEE (2022), additional unsupervised concepts are extracted by a self-explainable neural network (Alvarez-Melis and Jaakkola, NeurIPS (2018)), which guarantees in principle their interpretability. Since CBM-AUC is a natural competitor of DCBM, it should be clear why it is not considered in the empirical evaluation

In the experiments with CUB-20% and CUB-10%, it seems that any $\beta \leq 1 $ favors the intrinsic concepts to be highly informative of the label, refer to Fig. 7. Then, how can the interpretability of the predictions be guaranteed? For the results in Table 2, which $\beta$ do they refer to? This is not clear from the text, as in Sec. 5.2 the reader is referred to Appendix D, where there is only the study in relative performances upon changing $\beta$. In Appendix C.4 it is reported that: “the best model is chosen by either the best label accuracy or the best concept accuracy on validation set ”. Are those DCBMs with $\beta=0.1$ and $\beta=1$? Conversely, in the intervention experiment, $\beta=0.01$ and $\beta=0.1$ were used (I think there is a typo in the caption of Table 4, where there is $\beta=1.0$ but in the main text, it is said $\beta=0.1$). In this case, the implicit concepts are highly informative of the label, as clear from Figure 7.

Overall, it must be evaluated the relative importance of $\tilde f \circ \tilde g$ on the prediction w.r.t. $f \circ g$ to assess how the implicit concepts affect the downstream task. Without such a metric, interpretability cannot be guaranteed.

The proposal of the two-phase human-machine interaction is novel and interesting: it works effectively w.r.t. other interactive strategies, winning in all scenarios. Also, as a merit, it does not increase sensibly the computational cost. Here, a map $f_{DEC}$ is estimated to decouple the residual concepts from the explicit ones in the implicit layer. While I understood it is only used in the interactive section (correct me if I am wrong), it would be useful to adopt it also in the label prediction, since $\tilde c - f_{DEC} (\hat c)$ does not contain redundant information of the explicit concepts $\hat c$.


**Summary Of The Paper:**

In this paper, the authors introduce a new version of the Concept Bottleneck Model (Koh et al., 2020) that dims to be more accurate and interpretable w.r.t. other Concept-based models, especially in regimes where concept supervision is limited.
 The proposed method is inspired by a theorem depicting the trade-off between the concept error and the label error in the Joint Concept-Bottleneck model (CBM), due to missing concept information.  Hence, the authors propose a new version (DCBM) of the original model to decouple the information between explicit and implicit concepts. They conduct an extensive experimental comparison with other known CBMs and a standard black-box model, achieving overall better label/concept accuracy performances.
Along with a novel strategy to train the DCBM, they also propose a new two-stage human-machine interaction, addressing both the correction of the wrong concepts and of the misclassified labels. This proves effective and improves previous methods, without requiring costly time computations.



**Summary Of The Review:**

This paper presents a novel theoretical result (Theorem 1 for CBMs), which is relevant for future developments of interpretable concept-based models. Theorem 1 indicates an inherent trade-off between label and concept accuracy, up to some assumptions. However, it is not evident why the proposed model should address the limitations of CBMs, in light of the quantities introduced in the theorem. A decoupling phase is introduced in the interactive phase but is not present when giving the prediction, referring to Fig. 3.
In the experimental investigation, the proposed model (DCBM) achieves better downstream and interventional label/concept accuracy than CBM, especially in the regime of reduced-concept supervision.
Still, the claim of being interpretable is not fully supported by the experiments, whereas they indicate the implicit concepts (which are not interpretable) are highly predictive of the label in all regimes tested (exception made for $\beta=10$ in all datasets, Fig. 7).
Since the proposed model makes use of implicit concepts for the prediction, their relative importance has to be quantified somehow. Guaranteeing interpretability is necessary for DCBM, and for concept-based models in general, but it was just assumed by the authors. There is still the possibility that implicit concepts are not so relevant for the prediction, but it has to be shown. I am open to revising my score if this point is clarified.

---

> ### Author Response · Authors · 2022-11-10
> **Response to Reviewer k8KK (1/2)**
>
> Thank you for your time and valuable feedback, as well as your detailed comments and interest in our paper.
>
> According to your constructive comments, we make some replies to the questions:
>
> > The concern about assumption in the main Theorem.
>
> Thank you for your detailed discussion for our theoretical results. The Gaussian assumption of the weight matrix $W$ can be relaxed but the assumption $k<\min(d_1, d_2)$ is somehow not easy to improve. We have discussed our theoretical results in Appendix C.3 as follows:
>
> Note that we assume that the weight matrix $W_2 \in \mathbb{R}^{{d_2}\times k}$ is generated from a Gaussian distribution in Assumption 1, which is used when we bound the least singular value of the weight matrix. Note that the assumption can be relaxed as $rank(W_2)=k$ and we make such an assumption to obtain a more accurate description. Actually, for any matrix $W_2$, when we add a row $w\in \mathbb{R}^{k}$ to it, where $w$ can be regarded as the weight of a new concept, we will have $\sigma_{\min}(W_2) \leq \sigma_{\min}\left(\left[W_2;w\right]\right)$, which indicates that the decrease of concept number will increase the least singular value of the weight matrix, and hence increasing the label error according to Eq. 16.
>
> Moreover, we assume $k<\min (d_1,d_2)$, which is not consistent with the CUB dataset. This is one of the limitation of our theoretical result, and we regard this problem as an important future work.
>
> > The relationship between the main theorem and our methodology.
>
> On the one hand, Theorem 1 reveals that the weighted sum of the concept error ($\mathcal{E}_C$) and the label error ($\mathcal{E}_Y$) is bounded by $\Phi$. On the other hand, $\Phi(d_1, d_2)$ increases with more missing concepts $d_2$, thus the concept and label error of CBM will also increase correspondingly. Hence, Theorem 1 reveals that the concept/label trade-off will get worse with the missing of concepts. We have added this discussion to reveal the meaning of Theorem 1 in Sec. 2.2.
>
> The theorem and its proof reveal how the trade-off between concepts and labels occurs with insufficient concepts. According to our proof, CBM has to inject implicit information into the explicit one to achieve low label error, which will obviously sacrifice the concept accuracy. To decouple the aforementioned heterogeneous information, we devise an extra mapping $\tilde{f} \circ \tilde{g}$ to store the implicit information in the concept layer of the previous CBM. We have added this discussion in Sec. 3.1.
>
> > The interpretability of DCBM.
>
> Even with extremely insufficient concept information, DCBM can guarantee the relatively low concept/label error simultaneously, however, it is impossible to expect that all model decisions can be interpreted according to the limited concepts. Under this situation, we aim to make full use of the concept information and give interpretations for as many samples as possible. To this end, we utilize the JS Divergence as a metric to evaluate the interpretability for each data point, which reflects the change of DCBM's prediction caused by implicit concepts, i.e., the black-box parts.
>
> In short, DCBM and other models cannot give explanations for all data points with insufficient concepts. However, DCBM can give accurate predictions for data via explicit/implicit parts and meanwhile judge whether the model decision for this data point can be interpreted by given concepts via JS divergence. If the answer is yes, DCBM will give the corresponding interpretation.
>
> We have added this discussion in Sec. 3.1 and the corresponding experimental analysis in Sec. 4.4 and Appendix E.3.
>
> > $\beta$ in DCBM.
>
> According to your suggestions, we make the following revisions:
>
> (1) Typo: Yes, $\beta$ should be 0.01/1.0. We have revised it in our new version.
>
> (2) Which $\beta$ to choose: We illustrate which $\beta$ to use and the reason in Appendix D.4.
>
> (3) To better illustrate the impact of $\beta$, we implement lots of further experiments and analysis, including: a) In Appendix E.1 and Fig. 8(a), we illustrate how $\beta$ affects the values of JS Divergence; b) In Appendix E.3 and Fig. 9, we illustrate whether JS Divergence relates to interpretability; c) In Appendix E.4 and Fig. 10, we illustrate why the label accuracy of IMP-DCBM rises first and then falls with the increase of $\beta$.

---

> > ### Comment · Reviewer_k8KK · 2022-11-22
> > **Reply to the authors**
> >
> > Dear authors,
> > thank you for your time in replying to my and other reviews and addressing many points collected. The number of additional results is impressive and I revised my score for your paper.
> > I am happy with the clarifications that you provided in the comments and with the additional experiments, especially with the natural competitor CBM-AUC that I proposed. The revised results still indicate that DCBM owns better concept and label accuracy w.r.t. other state-of-the-art models.
> > A few comments are ahead:
> > 1) You use the JS divergence as a measure for the interpretability of the model. In particular, there are cases where a predicted class in CUB can have high JS divergence, thus relying mostly on implicit information. In this case, as stated by the authors, it is not possible to give an interpretation of the prediction in terms of explicit concepts. This eventually downgrades the model to a partially interpretable one, conflating with the claims of the main text. This is not a serious issue (but it must be recognized) as other concept-based models would not be interpretable either given a small number of known concepts.
> > 2) The results presented in figure 11 (a-b) indicate that the decoupled concepts contain some information about the explicit concepts (especially in CUB-10%) for high values of $\beta$. Hence, adopting high values $\beta$ may cause leakage of information of implicit concepts within the explicit learned. I would suggest that an optimal value of $\beta$ can be found upon further investigations, avoiding the concept leakage phenomenon. I acknowledge that this is somehow a separate problem from the purpose of the paper, but I think it would improve the manuscript and set the debate on interpretability.
> >
> > I believe that the results presented are promising and the proposal of decoupling explicit and implicit concepts remains a valid idea, but the manuscript, as it stands now, got heavier and some new parts are prone to discussion with other related works, like Concept Leakage.

---

> > > ### Author Response · Authors · 2022-11-30
> > > **Response to Reviewer k8KK**
> > >
> > > Thanks a lot for your positive comments and valuable suggestions on our work! We will attach importance to your suggestions and explore them in future work.

---

> ### Author Response · Authors · 2022-11-10
> **Response to Reviewer k8KK (2/2)**
>
> > The missing baseline CBM-AUC.
>
> Thanks for you to introduce this paper. We have carefully read this paper and cited it in the related work part. Unlike the JS Divergence in DCBM, CBM-AUC does not constrain the power of implicit knowledge, which makes it impossible to give faithful interpretation. However, we also decide to compare with it in these days and will update corresponding results before the deadline of revision phase.
>
> > Some comments on human-machine interactive. While I understood $f_{DEC}$ is only used in the interactive section (correct me if I am wrong), it would be useful to adopt it also in the label prediction, since does not contain redundant information of the explicit concepts.
>
> Yes, you are right. In the revised version, $f_{DEC}$ is only used in the interactive section. Thanks for your valuable suggestions. It is very interesting to adopt it in the label classification stage. We have conducted these experiments in Appendix E.6.

---

### Official Review · Reviewer_VESv · 2022-10-28

**Confidence:** 3
**Clarity, Quality, Novelty And Reproducibility:** Please see my comments above
**Correctness:** 2
**Technical Novelty And Significance:** 3
**Empirical Novelty And Significance:** Not applicable
**Recommendation:** 6

**Strength And Weaknesses:**

+The theoretical arguments about the tradeoff between concept prediction and label prediction accuracy when concepts are not sufficient to predict class labels are interesting and insightful.

+The paper focuses on decoupling implicit and explicit concepts which is an interesting solution to keep the explicit concept prediction accuracy intact and maintain high label accuracy.

+Maintaining high interpretability in CBMs is important and the authors have tried to preserve it by using a JS Divergence term.

+Leveraging mutual information minimization to enable forward and backward intervention on explicit concepts is an interesting approach that the authors have employed.

----------------
-Backward intervention is poorly described. Figure 4 ---which is about forward and backward intervention--- is not discussed in the text at all. The related equations are not properly described and are deferred to the appendix A but that appendix contain only a sudo-code with no additional explanation about the backward intervention formulation. What are the intuitions behind eq 4, 5 and 6?

-DCBM is claimed to be an interpretable model, but because the final decision is made based on both human-friendly concepts (explicit concepts) and implicit concepts (not interpretable), it seems that the entire model becomes not interpretable. Although JS Divergence loss term is applied to reduce the impact of implicit concepts on the prediction, the final decision still depends on both implicit and explicit concepts.

-It was mentioned that joint concept/label training in CBM suffers from concept distortion and the concept decoupling model is proposed to mitigate this problem. When the weight of JS Divergence goes to infinity, DCBM converges to CBM. When using less number of explicit concepts, it is expected that the label accuracy of CBM (as well as DCBM with large JS weight) decreases. But we don’t see such a pattern in Figure 7. Looking at sub-figures a-d they all have a similar label accuracy of around 80%. This is counterintuitive. According to Figure 7, CUB%10 and CUB have almost similar CBM label accuracy. Same for Exp-CBM with the largest JS weight.

-A similar counter-intuitive observation can be made in Table 3. Label accuracy reported for CUB, CUB%50, CUB%20 and CUB%10 all are in the range of 80%. I expected that all methods perform much worse on CUB%10 dataset. How this similar performance can be justified?


-Another counterintuitive behavior in Figure 7 is about the label accuracy for IMP-DCBM. My understanding is that as the JS weight increases, implicit concepts become less informative about class labels (harsher decoupling) and the label accuracy of IMP-DCBM should decrease. But we don’t see such a pattern in any of the sub-figures of Figure 7. In Figure 7, by increasing JS weight, label accuracy of IMP-DCBM increases and then decreases.  How the initial increase can be justified?

-How can we guarantee that the backward intervention captures wrong predicted concepts and not correct concepts?

-What are the exact loss terms used as L_Y and L_C. Are they conventional cross-entropy?

-In Table 4 that the concept accuracy on samples with wrong label prediction is reported, what is the proportion of the misclassified samples that the authors do intervention on? Did you rectify all the samples with a wrong assigned label?
-It would be interesting to have an experiment to show how well the decoupling network is able to reduce the degree of mutual information when different number of explicit concepts are used.


**Summary Of The Paper:**

Concept Bottleneck Model (CBM) has the advantage of interpretability. One of the issues that limits their use is insufficiency of high-level concepts, that is, available concepts (explicit concepts) do not encode sufficient information to predict class labels. Hence, the paper suggested to use additional concepts (implicit concepts) which are essentially additional neurons in the bottleneck layer. Implicit concepts are introduced to avoid pollution of explicit concepts during training. The paper proposes Decoupling Concept Bottleneck Model (DCBM) which is a concept-based model decoupling heterogeneous information into explicit and implicit concepts. Moreover, they proposed two methods (forward intervention and backward rectification,) which can automatically correct labels and trace back to wrong concepts.


**Summary Of The Review:**

It seems that there are interesting ideas but not properly described. More importantly, I have major concerns about the experiments and I hope the authors can clarify.

---

> ### Author Response · Authors · 2022-11-10
> **Response to Reviewer VESv (1/2)**
>
> Thank you for your time and valuable feedback, as well as the positive comments and interest in our paper.
>
> According to your constructive comments, we make some replies to the questions:
>
> > The intuitions behind equations in the human-machine interactive section.
>
> We feel sorry that the original paper is not clear enough in the human-machine interaction section. To make the methodology part more readable and take the paper length into account, we introduce the high-level idea in the main part (Sec. 3.2) and the detailed algorithm in Appendix B. We refresh this part and introduce the intuitions behind the formulas. We hope the new version can meet your requirements.
>
> > The interpretability of DCBM.
>
> Thanks for your valuable comments. We have discussed and analyzed the interpretability of DCBM in the revised version as follows:
>
> Even with extremely insufficient concept information, DCBM can guarantee the relatively low concept/label error simultaneously, however, it is impossible to expect that all model decisions can be interpreted according to the limited concepts. Under this situation, we aim to make full use of the concept information and give interpretations for as many samples as possible. To this end, we utilize the JS Divergence as a metric to evaluate the interpretability for each data point, which reflects the change of DCBM's prediction caused by implicit concepts, i.e., the black-box parts.
>
> In short, DCBM and other models cannot give explanations for all data points with insufficient concepts. However, DCBM can give accurate predictions for data via explicit/implicit parts and meanwhile judge whether the model decision for this data point can be interpreted by given concepts via JS divergence. If the answer is yes, DCBM will give the corresponding interpretation.
>
> We have added this discussion in Sec. 3.1 and the corresponding experimental analysis in Sec. 4.4 and Appendix E.3.
>
> > The counter-intuitive result in Fig. 7 (Fig. 6 in the revised version). According to Fig. 7, CUB%10 and CUB have almost similar CBM label accuracy.
>
> This intuition is definitely correct, but we feel sorry that the figure in the original version is easy to be misunderstood. In fact, there is a relatively large gap between the label accuracy of CUB and CUB%10, which are 79.80% and 75.09%, respectively. However, both of them seem to be around 80% due to the different scales of the y-axis in the original figure. To this end, we unified the scales of the y-axis in the new version. Furthermore, we add a dotted line in Fig. 6 (a-e) to express the label accuracy of Joint-0.01, and it can be seen that the performance of DCBM convergences to CBM when the $\beta$ is sufficiently large ($\beta=10$).
>
> Thanks for your comments on improving the clarity of our paper.
>
> > The counter-intuitive observation can be made in Table 3 (Table 2 in the revised version). Label accuracy reported for all datesets are in the range of 80%. I expected that all methods perform much worse on CUB%10 dataset.
>
> Actually, take $\beta=1.0$ as an example, the label accuracy of DCBM-0.1 in Table 3 is the same as that in Table 1. (In the revised version, all $\beta$s for DCBM-0.1 are the same as 1.0) These results show a counterintuitive but interesting phenomenon that label accuracy even increases with higher ratios of concepts discarded on CUB, which is contrary to any other approach. To illustrate this, we together consider the label accuracy of DCBM-0.1, DCBM-0.01, and _Standard_ model without bottleneck (abbreviated as STAN). It is intuitive that with more ratios of concepts discarded on CUB, both DCBM-0.1 and DCBM-0.01 will lose interpretability and converge to STAN, no wonder the label accuracy of DCBM-0.1 and DCBM-0.01 converges to the one of STAN, which is 82.54%. Therefore, the reason in our conjecture that the label accuracy of DCBM-0.1 decreases is because $\alpha=0.1$ is so large that it forces the model to learn concepts better while, to some extent, affecting the label accuracy. However, for DCBM-0.01, $\alpha=0.01$ is small and under this situation, the concepts can promote label accuracy.
>
> We add this analysis in Sec. 4.3.

---

> > ### Comment · Reviewer_VESv · 2022-11-17
> > **Accuracy after forward intervention**
> >
> > In Table 3 about Forward intervention, for CUB%10 why the label accuracy of the JOINT model has dropped (75.53 --> 67.85) after intervention? Shouldn't intervention always improve the accuracy? The accuracy drop is rather large. Same for DERM7PT (83.02 --> 74.15)

---

> > > ### Author Response · Authors · 2022-11-17
> > > **Response to Reviewer VESv**
> > >
> > > Thanks for your reply. Actually, intervention cannot always improve label accuracy. This phenomenon can also be observed in the original CBM paper [1] in their Figure 4 (left and middle). As discussed in [1], one major reason for this phenomenon is that when the concept accuracy is low, the learned concepts are not necessarily aligned with the ground truth concepts. While in our paper, we find it more remarkable when concepts are insufficient, so the more inherent reason is mainly due to the concept/label distortions discussed in our paper.
> > >
> > > In the cases CUB%10 and Derm7pt, concepts are severely insufficient, so _Joint_ model, to some extent, sacrifices concept accuracy to get a higher label accuracy. Under this perspective, the intervention in concepts on the contrary invalidates the sacrifice and harms the label accuracy.
> > >
> > > We add this analysis in Sec. 5.
> > >
> > > [1] Koh, Pang Wei, et al. "Concept bottleneck models." International Conference on Machine Learning. PMLR, 2020.

---

> > > > ### Comment · Reviewer_VESv · 2022-11-17
> > > > **Real-world implication of the above phenomenon**
> > > >
> > > > Q1: Just to be clear, does the concept bottleneck layer (g) pass binary concept predictions (absence/presence of a concept) or soft scores (between 0 and 1) to f?
> > > >
> > > > Q2: The scenario you described above seems to be like this: We have a model that makes right decisions based on wrong concepts predictions and when a human expert performs forward intervention on concepts, the label predictions become wrong. right?
> > > >
> > > > Q3: In a hypothetical scenario, if you train the concept prediction network g independently and freeze g and only train f to predict class labels, then do you think forward intervention would always have to improve label accuracy?
> > > >
> > > > Q4: Out of the above questions, I am trying to understand the real-world implication of the above phenomenon (the one I described in Q2). One possibility is this: we have a concept-based model that has a good accuracy but it is not interpretable because it behaves counter-intuitive during forward intervention. Effectively it is more like a black-box model although it has a concept layer. But, we can come to the same conclusion by simply examining concept accuracies. A poor concept accuracy followed by a high class label accuracy can indicate that the model is effectively a black-box model in spite of having a bottleneck layer. Therefore, a forward intervention is not needed to come to such a conclusion. Do you agree?

---

> > > > > ### Author Response · Authors · 2022-11-17
> > > > > **Response about Real-world Implication**
> > > > >
> > > > > Thanks for your reply.
> > > > >
> > > > > A1: The concept layer outputs soft scores to the classifier $f$.
> > > > >
> > > > > A2: Yes, this is exactly what we want to say.
> > > > >
> > > > > A3: The model you described in Q3 is the same as the _Sequential_ model described in the original CBM paper [1] and we have summarized it in Appendix A. According to their experiments (Figure 4 in [1]), the forward intervention can always improve label accuracy.
> > > > >
> > > > > A4: Yes, we agree with you. When the concept accuracy is relatively low, the concept layer will be mixed with black-box information. Under this situation, CBM will typically lose interpretability, which makes forward intervention meaningless.
> > > > >
> > > > > [1] Koh, Pang Wei, et al. "Concept bottleneck models." International Conference on Machine Learning. PMLR, 2020.

---

> > > > > > ### Comment · Reviewer_VESv · 2022-11-17
> > > > > > **forward intervention and label accuracy**
> > > > > >
> > > > > > g outputs soft scores. f expects soft scores. But during forward intervention, we replace the soft scores of the intervened concepts with ground-truth values, which are binary values (hard scores). Do you think this could be a source of performance decrease due to intervention? If, yes, even in the Sequential model, just replacing soft scores with ground-truth hard scores can decrease the label accuracy. Right?

---

> > > > > > > ### Author Response · Authors · 2022-11-18
> > > > > > > **Response about forward intervention**
> > > > > > >
> > > > > > > Thanks for your question!
> > > > > > >
> > > > > > > Actually, both CBM and DCBM do not replace soft scores with ground-truth hard scores directly. As discussed in Sec.5: _Concept values are respectively intervened to the 95th or 5th percentile of the ones over the training distribution for positive/negative samples, which is in line with (Koh et al., 2020)._

---

> ### Author Response · Authors · 2022-11-10
> **Response to Reviewer VESv (2/2)**
>
> > Another counter-intuitive behavior in Fig. 7 (Fig. 6 in the revised version). In Fig. 7, by increasing JS weight, label accuracy of IMP-DCBM increases and then decreases. How the initial increase can be justified?
>
> In the loss function of DCBM, $L_1$ constrains model output, i.e., the sum of explicit and implicit outputs to close to the ground-truth label, and $L_3$ constrains the JS Divergence of the model and explicit outputs. We only regard it as a supplement for the explicit one and there is no guarantee for the performance of IMP-DCBM in the loss function. However, it's still interesting why the label accuracy of IMP-DCBM rises first and then falls on CUB.
>
> Actually, by increasing $\beta$, the distribution of explicit outputs can be forced to be similar to the sum of explicit and implicit outputs, which leads to two consequences: 1) the implicit outputs become similar to the explicit ones; 2) the implicit outputs tend to 0. Under this perspective, we use the JS Divergence of explicit and implicit outputs in Fig. 10 (a) to illustrate the former and the MAV of implicit outputs in Fig. 10 (b) to illustrate the latter. Looking first of all at CUB in Fig. 10 (a), the JS Divergence keeps stable from 0.0 to 0.1, so it's reasonable for the label accuracy of IMP-DCBM to rise together with EXP-DCBM, but it finally drops because the JS Divergence surges. By comparison, there's little rise in the label accuracy of IMP-DCBM when $\beta$ changes from 0.0 to 0.1 on CUB%10, though an obvious decline of JS Divergence exists on CUB%10. The reason is that due to the low accuracy of EXP-DCBM, the approximation to explicit outputs cannot result in a better label accuracy for IMP-DCBM. These observations are consistent with that shown in Fig. 6. Moreover, in Fig. 10 (b), the mean absolute value (MAV) of implicit concepts also converges to 0 as expected.
>
> We add the above illustration in Appendix E.4.
>
> > How can we guarantee that the backward intervention captures wrong predicted concepts and not correct concepts?
>
> In the forward intervention task (Koh et al. 2020), the main idea is to correct the label according to the ground-truth concepts given by experts. Inspired by this task, we believe that a more accurate label will also help the identification of misclassified concepts, and thus, we propose backward rectification, a novel human-machine interactive task, in this paper. In fact, the backward rectification task is very difficult because the ratio of misclassified concepts is very low, which is less than 4% in general. Thus, our original intention is to choose several predicted concepts which are possibly wrong and then query them to the human experts. Moreover, we display the true positive rate (TPR) and false positive rate (FPR) in Table 3 to address your concern.
>
>
> > What are the exact loss terms used as $L_Y$ and $L_C$. Are they conventional cross-entropy?
>
> We use cross-entropy loss during the training stage, and have introduced the loss function in Appendix D.3.
>
> > Did you rectify all the samples with a wrong assigned label?
>
> Yes, we rectify all samples with the wrong assigned labels in the backward rectification experiments.
>
> > It would be interesting to have an experiment to show how well the decoupling network is able to reduce the degree of mutual information when different number of explicit concepts are used.
>
> Thanks for your valuable suggestion! It is very interesting to study the effects of decoupling netwrok. We have conducted these experiments in Appendix E.6.

---

> ### Author Response · Authors · 2022-11-30
> **A kind Reminder of the Post-rebuttal Feedback**
>
> Dear Reviewer VESv,
>
> We greatly appreciate your valuable and constructive comments. We would like to know whether there are some other questions about our work? We are happy to answer and discuss them. If your questions have been addressed, could you please kindly raise the rating?
>
> Sincerely,
>
> Paper 333 Authors

---

### Official Review · Reviewer_ZauB · 2022-10-29

**Confidence:** 4
**Correctness:** 3
**Technical Novelty And Significance:** 2
**Empirical Novelty And Significance:** 3
**Recommendation:** 5

**Clarity, Quality, Novelty And Reproducibility:**

Clarity: Discussed above.
Quality: Adequate
Novelty: Adequate
Reproducibility: Adequate but there was no code release for review.

**Strength And Weaknesses:**

Strengths
- The paper's presentation is strong. Figures are immaculate!
- The method addresses serious issues with CBMs. I would have preferred a strong motivating (running) example that naturally suggests the need for implicit and explicit concepts.
- The theory seems correct; however, the authors could do a much better job at motivating its need (perhaps using the aforementioned example). At the moment, it is unclear how the proposed theory impacts the subsequent algorithms/experiments.
- The notion of forward intervention and backward rectification are sensible and interesting.

Weaknesses
- While the paper is quite a whirlwind and looks thorough, there are some clarity issues. Instead of easing readers into a clear, convincing narrative, the authors elected to inundate the readers with results without much interpretation.
- There is a nomenclature issue with Section 6. It's a stretch to call the proposed system as a human-machine system. There is no human subject experiment to validate this. It's unclear if humans will intervene and rectify sensibly. The experiments presume orcale-esque behavior from the human. Instead of walking through the results, the authors make it difficult to follow the results and their implications. Supporting prose can be of much help to all.

**Summary Of The Paper:**

This is a strong paper that proposes methods that right issues arising from concept bottleneck models. The authors show how to split concepts into explicit and implicit information, and then demonstrate how to debug concept and label errors.

**Summary Of The Review:**

This was a very interesting paper that clearly has legs to improve CBMs. However, the lack of clarity in message makes it hard to accept as is. The density of the paper could be reduced to provide the readers clear motivation for the proposed methods.

---

> ### Author Response · Authors · 2022-11-10
> **Response to Reviewer ZauB**
>
> Thank you for your time and valuable feedback, as well as your positive comments and interest in our paper.
>
> According to your constructive comments, we make some replies to the questions:
>
> > How the proposed theory impacts the subsequent algorithms/experiments?
>
> The theorem and its proof reveal how the trade-off between concepts and labels occurs with insufficient concepts. According to our proof, CBM has to inject implicit information into the explicit one to achieve low label error, which will obviously sacrifice the concept accuracy. To decouple the aforementioned heterogeneous information, we devise an extra mapping $\tilde{f} \circ \tilde{g}$ to store the implicit information in the concept layer of the previous CBM. We have added this discussion in Sec. 3.1.
>
> > While the paper is quite a whirlwind and looks thorough, there are some clarity issues. Instead of easing readers into a clear, convincing narrative, the authors elected to inundate the readers with results without much interpretation.
>
> We feel sorry that the paper is not clear enough in the original version, and we have revised our paper in the new version. The introduction of all revisions can be seen in the replies for all reviewers. Following your suggestion, we refresh the human-machine interactive part (Sec. 3.2 and Appendix B) and introduce the intuitions behind the formulas. To make the methodology part more readable and take the paper length into account, we introduce the high-level idea in the main part (Sec. 3.2) and the detailed algorithm in Appendix B.
>
>
> > There is a nomenclature issue with Section 6. It's a stretch to call the proposed system as a human-machine system. There is no human subject experiment to validate this. It's unclear if humans will intervene and rectify sensibly. The experiments presume orcale-esque behavior from the human. Instead of walking through the results, the authors make it difficult to follow the results and their implications. Supporting prose can be of much help to all.
>
> Thanks for your valuable comments and suggestions. In this paper, we do not interact with real-world human experts in the experiments but assume that the judgments given by human experts are consistent with the concept/label values in the dataset. Therefore, we only propose a feasible plan of human-machine interaction via DCBM and it requires further examination to demonstrate whether the system can work as expected. We have discussed your concern at the beginning of Sec. 5 and added further analysis in the experimental parts (Sec. 5, Appendix E. 5 and E. 6).

---

> ### Author Response · Authors · 2022-11-30
> **A Kind Reminder of the Post-rebuttal Feedback**
>
> Dear Reviewer ZauB,
>
> We greatly appreciate your valuable and constructive comments. Your main concern in the initial review seems to be the clarity of the original paper. Accordingly, we have taken great efforts to revise the paper. We would like to know whether there are some other questions about our work? We are happy to answer and discuss them. If your questions have been addressed, could you please kindly raise the rating?
>
> Sincerely,
>
> Paper 333 Authors

---

> ### Author Response · Authors · 2022-12-07
> **A Second Reminder of the Post-rebuttal Feedback**
>
> Dear reviewer ZauB,
>
> Thanks again for your time for reviewing our paper and the updated version. From the positive feedback with increased scores, the new version with improvement in presentation and more experimental results may have addressed most concerns from the other three reviewers (Reviewer k8KK, Reviewer VESv, and Reviewer FEGm). Could you take a look and reconsider your rating? Thank you.
>
> Sincerely,
>
> Paper 333 Authors

---

### Official Review · Reviewer_hv7P · 2022-10-30

**Confidence:** 5
**Correctness:** 2
**Technical Novelty And Significance:** 2
**Empirical Novelty And Significance:** 2
**Recommendation:** 3

**Clarity, Quality, Novelty And Reproducibility:**

The work is well written but parts are incomplete in the sense that several crucial comparisons are missing and as a result the work doesnt feel as novel.



**Strength And Weaknesses:**

Overall the work addresses a very relevant problem and presents a good solution to the proposed issue. The method is well motivated and the authors make several comparisons on CUB and DERM7PT data sets.

There are definitely several shortcomings though. Perhaps the most critical of these is the fact that the authors are missing several crucial citations that can potentially impact the quality of the analysis and significance of the results. In particular, the authors never mention works such as Mahinpei et al 2021 (https://arxiv.org/pdf/2106.13314.pdf) who describe some of the major pitfalls of CBMs. Mahinpei et al 2021 explicitly talk about this unexpected behaviour of concept bottlenecks in terms of leakage and several followup works from the same group e.g Havasi et al 2022 present some solutions with a side channel using mutual information to separate the concept information into two subsets to overcome this leakage issue.  I would like to see some comparisons here as to how does the issue of having insufficient concept information relate to leakage? And similarly, how does the approach compare against Havasi et al 2022.

Mahinpei et al show that while Concept models encourage concept dimensions to be uncorrelated, this does not completely
prevent information leakage – each concept dimension can still encode for multiple concepts and that the
concept dimensions can nonetheless be statistically dependent. They state that CBM training should explicitly
minimize mutual information between concept dimensions – both aligned and unaligned - if concepts are believed to be independent. Similar views are presented in Klys et al 2018. How does the proposed method compare to these suggestions?

Finally, I would also like to see some discussion and analysis on how having the human-in-the-loop affects the concept definitions. Is it expected that the human intervenes on the concepts akin to interventions in the causal sense?



**Summary Of The Paper:**

The paper examines concept bottleneck models and focuses on the case where these models have unexpected behaviour when there is insufficient concept information. The authors propose a strategy for decoupling CBMs into explicit and implicit concepts while retaining high predictive performance and interpretability. They devise two algorithms based on mutual information to automatically correct labels and trace back to incorrect concepts. Based on this a human may intervene to correct incorrect concept definitions.

**Summary Of The Review:**

Overall the paper is well written but the experimental section needs a lot of work in terms of comparing with existing works tackling similar issues and a significantly better job needs to be done in terms of contextualising the contributions of the paper compared to existing works.

The work seems to lack novelty for the most part.

---

> ### Author Response · Authors · 2022-11-10
> **Response to Reviewer hv7P**
>
> Thank you for your time and valuable feedback.
>
> To begin with, we have conscientiously read the paper you list, some of which are what we have read before; however, an exception is "Havasi et al 2022" - whether it represents "Addressing leakage in concept bottleneck models"? Actually, we noticed its title in the acceptance list of NeurIPS22 before submission, but we failed to find a pre-print version at that time. To our knowledge, it was available online a month after the submission deadline of ICLR 2023.
>
> We understand that your main concern is about leakage, which is indeed a shortcoming in most CBM-based models; however, CBM still has basic values and inspires a series of subsequent research work and novel thinking, so we guess research without targeting on leakage can still have research value.
>
> Though we acknowledge that leakage is an important and interesting issue in CBMs, we have to claim that our motivation is to alleviate the concept/label distortions when concepts are insufficient. We guess its value must be underestimated while we are the first to give theoretical proof of the distortions.
>
> Leakage may of course exist in DCBM, as it does not solve the disadvantages of CBMs under this perspective. Further research on leakage might be interesting and meaningful but after careful consideration, we regard it as our future work because it's another large topic apart from distortions. Moreover, compared with leakage, distortions can more significantly affect the predictions, so it's intuitive to use concept/label accuracy to evaluate the concept/label distortions while for leakage, some delicate metrics should be further devised.
>
> According to your suggestions, we display the relationship and difference between leakage and distortions in the part of related work.
>
> We hope the reviewer could pay attention to our contribution to concept/label distortion and its theoretical proof, which we believe has basic value.
>
> > I would also like to see some discussion and analysis on how having the human-in-the-loop affects the concept definitions. Is it expected that the human intervenes on the concepts akin to interventions in the causal sense?
>
> In this paper, we basically assume that concepts are well defined by human experts, which is the same setting as CBM and thus, these concepts are not supported to be redefined or extracted by human-in-the-loop mechanism like [1].
>
> Furthermore, we discussed and cited several papers about causal effect of CBM during the intervention phase or model the causal relationship between concepts in the related work parts (Appendix A). However, in this paper, we do not consider the causal effect and regard it as an important future work.
>
> [1] Z. Zhao, P. Xu, C. Scheidegger and L. Ren, "Human-in-the-loop Extraction of Interpretable Concepts in Deep Learning Models," in IEEE Transactions on Visualization and Computer Graphics, vol. 28, no. 1, pp. 780-790, Jan. 2022, doi: 10.1109/TVCG.2021.3114837.

---

> ### Author Response · Authors · 2022-11-30
> **A kind Reminder of the Post-rebuttal Feedback**
>
> Dear Reviewer hv7P,
>
> We greatly appreciate your valuable and constructive comments. We acknowledge that leakage is a very important issue in CBM; however, since our manuscript has already been very heavy now, we only consider leakage as future work, but we still attach importance to your suggestions. Specifically, we cite all the papers about leakage you referred, introduce them in the part of related work with regard to leakage, and highlight it as a limitation and future work in the part of conclusion. We are happy to have further discussions if you have other questions.
>
> Sincerely,
>
> Paper 333 Authors

---

### Official Review · Reviewer_FEGm · 2022-11-01

**Confidence:** 3
**Correctness:** 3
**Technical Novelty And Significance:** 3
**Empirical Novelty And Significance:** 3
**Recommendation:** 8

**Clarity, Quality, Novelty And Reproducibility:**

All sections except the results one are perfectly clear and well-written. The result section would gain a bit of clarity by moving certain details to the appendix while bringing ablation on $\beta$ to the main paper. This work shares ideas with a concurrent work (see comment above), but is of a higher quality and additionally addresses shortcut learning problems with implicit concepts. Thus, I believe the idea to be novel. The problem and conducted experiments are well-motivated but providing extended details about the experimental setting and baseline is needed. Reproducibility could be improved by providing code.

**Strength And Weaknesses:**

Strengths:
- The problem of performance decrease with the number of concepts is highly relevant to the field of interpretability given CBM's popularity.
- The authors have valuable experiments and theoretical findings supporting the existence of limitations for CBM in low concept regime
- The method and metrics are well explained making the paper easy to follow.
- Contrary to other works, the authors consider the possible issue of shortcut learning through implicit concepts and propose a solution for it.
- While their method is clearly better in all classification aspects compared to CBM, the authors also perform an analysis of the computational overhead.
- Authors provide experiments on the impact of both introduced parameters, JSD weight $\beta$ and concept-label trade-off term $\alpha$. (but in the appendix, see below)

Weaknesses:
- The experiments section is sometimes hard to follow, there are some key aspects of the experiment that are unclear to me:
  -  What is the value of $\beta$ in Table 2 for DCBM ?
  - How did the authors select $\tilde{d}$, the implicit concept dimension ? What is the impact of this parameter on performance?
  - What do the authors mean by "computational overhead" in table 5, is it compared to CBM alone? If yes, what is the relative comparison? Is this time per epoch or over an entire training process?
- The authors did not provide code yet which forbids any conclusion with respect to reproducibility.
- The impact of JSD weight $\beta$ is core to this work, as it's the only way to avoid shortcut learning through implicit concepts. Thus the content of figure 7 belongs in the main paper and should be further discussed.
- Results from Figure 7 (c,d) and Table 3 seem to be inconsistent. Indeed in Figure 7, for CUB 10% and 20% DCBM($\beta$ = 0.01) relies mainly on implicit concepts with an accuracy below 50% using explicit concepts only. However, in Table 3, forward intervention (which only affects explicit concepts) has a greater impact than for CUB and CUB 50% where DCBM($\beta$ = 0.01) does achieve good classification accuracy from explicit concepts. The authors never discuss this matter. What are the underlying reasons?


This is not a weakness given the recentness of the work, but I believe authors should consider citing "Clinical outcome prediction under hypothetical interventions – a representation learning framework for counterfactual reasoning" (2022) that propose partial concept bottleneck (PCB). They also propose to learn another discrete representation to alleviate the lack of concepts. However, their work is more limited and does not consider the crucial issue of shortcut learning from implicit concepts making interpretability impossible.

**Summary Of The Paper:**

In this work, authors analyze the current trade-off between label and concept accuracy in concept bottleneck models when the number of concepts reduces. They prove this trade-off to be inherent to standard CBM  models both theoretically and empirically. Based on their observation, they propose a new approach named DCBM, which allows the model to learn additional "implicit" concepts to improve label classification performance when concepts are missing while ensuring good concept classification. To avoid models to bypass explicit concepts in the decision-making of their downstream task, the authors also ensure low JS divergence between explicit concept predictions and label predictions. They show DCBM superiority in a low concept regime over previous CBM models on two benchmark datasets. They also show that their model performs well under forward intervention and their newly introduced backward rectification task.

**Summary Of The Review:**

Overall the authors propose a simple and efficient method for a highly relevant problem in the field. Their work is clearly structured and written. Their experiments show strong results and are well-motivated. However, crucial details surrounding these experiments and discussions are missing to back the authors' claims, in particular around the JSD weighting parameter $\beta$ and its impact.

---

> ### Author Response · Authors · 2022-11-10
> **Response to Reviewer FEGm**
>
> Thank you for your time and valuable feedback, as well as the positive comments on our motivation, experiments, and theories.
>
> According to your constructive comments, we make some replies to the questions:
>
> > What is the value of $\beta$ in Table 2 (Table 1 in the revised version) for DCBM ?
>
> We details the $\beta$ in Appendix D.4 in the revised version. In Table 1, we display the results of DCBM-0.1 and DCBM-0.01 using $\beta=1.0$ on almost all datasets. The mere exception is DCBM-0.01 on CUB, where we use $\beta=0.1$ because concepts are sufficient, and we don't need a large $\beta$ to restrict JS Divergence. Such settings can well balance the model's accuracy and interpretability, as detailed in Fig. 6.
>
> > How did the authors select $\tilde{d}$, the implicit concept dimension ? What is the impact of this parameter on performance?
>
> We supplement the effects of $\tilde{d}$ of [32, 64, 128, 256, 512] in Fig. 8 (b-c) with $\alpha=0.1$ and $\beta=0.01$ and it can be concluded that $\tilde{d}$ has few impacts on concept/label accuracy.
>
> > What do the authors mean by "computational overhead" in Table 5, is it compared to CBM alone? If yes, what is the relative comparison? Is this time per epoch or over an entire training process?
>
> In Table 5, we don't compare computational overhead to CBM, but compare different processes in DCBM, including model training, decoupling neural networks, forward intervention, and backward rectification. Our aim is to illustrate that the forward intervention and backward rectification processes, based on a light min-max optimization, are much faster and more efficient than model training. In addition, the time shown in this table is not per epoch, but over an entire training process. To better illustrate this, we make some corresponding revisions in the new version.
>
> > The authors did not provide code yet which forbids any conclusion with respect to reproducibility.
>
> We will submit the code in supplementary materials before the deadline of revision phase.
>
> > The impact of JSD weight $\beta$ is core to this work, as it's the only way to avoid shortcut learning through implicit concepts. Thus the content of figure 7 (It's Fig. 6 in the revised version) belongs in the main paper and should be further discussed.
>
> We move the section of JSD weight discussion, together with Fig. 6, to the main text, as it's indeed important in this paper. Moreover, to better illustrate the impact of $\beta$, we implement lots of further experiments and analysis, including: a) In Appendix E.1 and Fig. 8(a), we illustrate how $\beta$ affects the values of JS Divergence; b) In Appendix E.3 and Fig. 9, we illustrate whether JS Divergence relates to interpretability; c) In Appendix E.4 and Fig. 10, we illustrate why the label accuracy of IMP-DCBM rises first and then falls with the increase of $\beta$.
>
> > Results from Fig. 7 (c,d) and Table 3 seem to be inconsistent.
>
> Your concern is interesting. It's intuitive that DCBM-REC and DCBM-REC-DEC are supposed to have a greater impact on CUB and CUB%50 than CUB%20 and CUB%10 with $\beta=0.01$ because CUB and CUB%50 rely more on explicit concepts; however, the truth is opposite. We argue that the underlying reasons may be as follows:
>
> (1) We respectively intervene 11 (10% of the total 112) concept values for wrong predictions on CUB and its variants. Therefore, we intervene 100% concepts on CUB%10 and 50% concepts on CUB%20. This is the main reason why the prediction promotion is greater on CUB%20 and CUB%10 than CUB and CUB%50.
>
> (2) Though the label accuracy predicted only by explicit concepts is very low on CUB%20 and CUB%10 with $\beta=0.01$, explicit concepts do affect the comprehensive prediction much. For instance, in Fig. 6 (d), the label accuracy of IMP-DCBM is 75.74%, but with the help of EXP-DCBM, whose label accuracy is only 14.73%, the label accuracy of DCBM can hit a high of 82.45%. Thus, we can conclude that explicit concepts and the label accuracy of DCBM have a strong relation even though the label accuracy of EXP-DCBM is inappreciable.
>
> (3) Followed by (2), DCBM has a lower concept accuracy on CUB%20 and CUB%10, as shown in Table 1, so it's intuitively reasonable that the promotion on CUB%20 and CUB%10 has a larger room for improvement. Actually, this is not a sole phenomenon. Similar results exist that _Joint_ model promotes 3.35% label accuracy on CUB%50, which is higher than 2.4% on CUB. Besides, _CME_ promotes label accuracy by 4.1%, 6.06%, 9.79%, and 16.39% on CUB, CUB%50, CUB%20, and CUB%10, respectively, which also shows this observation.
>
> To avoid misunderstanding and illustrate the underlying cause, we refresh the writing around forward intervention in Sec. 5.
>
> > I believe authors should consider citing "Clinical outcome prediction under hypothetical interventions – a representation learning framework for counterfactual reasoning" (2022).
>
> Thanks for your advice. We have cited it in the new version.

---

> > ### Comment · Reviewer_FEGm · 2022-11-11
> > **Answer to authors post-rebuttal**
> >
> > I thank the authors for their clarifications.
> >
> > They have addressed all the concerns I had and in particular, they clarified the counter-intuitive results in the forward intervention. From my understanding, I thought the number of corrected concepts was proportional and not absolute but this is clear now.
> >
> > The fact that the authors used  $\beta=1.0$ for the main experiment confirms that higher accuracy can be achieved with DCBM while preserving a high dependence on explicit concepts. This is also confirmed by the forward intervention results.
> >
> > Overall I believe the revised manuscript to be of higher quality. Thus, I'll raise my score.
> >
> > I also went through other reviews which raised other questions. If the authors believe to have addressed them, as I'm less familiar with the affected aspects of the work, I'll keep the same confidence in my review.

---

> > > ### Author Response · Authors · 2022-11-17
> > > **Response to Reviewer FEGm**
> > >
> > > Thanks a lot for your positive comments and valuable suggestions on our work!

---

### Author Response · Authors · 2022-11-10
**General Response**

Dear Area Chairs and Reviewers,

We appreciate the reviewers’ time, valuable comments and constructive suggestions. From an overall perspective, the reviewers regard our paper well-written (FEGm, hv7P, k8KK), well-motivated (FEGm, hv7P, ZauB), and interesting (ZauB, VESv, k8KK), and meanwhile, the problem of the concept/label distortion is serious (ZauB), important (VESv, k8KK), and highly relevant (FEGm, hv7P) to interpretability. In addition, we are glad to see that reviewers FEGm, ZauB, VESv, and k8KK approve the originality of our theoretical contributions.

However, some expressions may, to some extent, mislead the readers, so we make some revisions to clarify some details in the revised version, and as such, reviewers can better understand and evaluate our contributions. We believe that the concerns from reviewers are not fatal to this paper and can be solved in the revised version.

The major revisions are listed as follows:

1. We explain the meaning of the main theorem in Sec. 2.2, and introduce the relationship between the main theorem and our methodology in Sec. 3.1, i.e., how is DCBM motivated by the theoretical result? Furthermore, we relax the Gaussian distribution assumption and discuss the limitation of our theoretical result in Appendix C.3.

2. We add a paragraph to discuss the interpretability of DCBM in Sec. 3.1. Furthermore, we conduct additional experiments to demonstrate the relationship between interpretability and JS Divergence in Appendix E.3.

3. We refresh the human-machine interactive part (Sec. 3.2 and Appendix B), and introduce the intuitions behind the formulas. To make the methodology part more readable and take the length of paper into account, we introduce the high-level idea in the main part (Sec. 3.2) and the detailed algorithm in Appendix B.

4. We add several related works in Appendix A, including the leakage in CBMs, causal effects of CBM, and other CBMs using implicit information.

5. We discuss some interesting but counterintuitive experimental phenomena of our results in Sec. 4.3.

6. We display the relationship between the weight and values of JS Divergence in Appendix E.1 and how much the dimension of implicit concepts affects concept/label accuracy in Appendix E.2.

7. We study the properties of decoupling network, including how well the decoupling network is able to reduce the MI in Appendix E.5, and its effects during prediction in Appendix E.6.

Also, we are trying our best to implement other results or analysis suggested by reviewers, including:

1. Compare with an additional baseline CBM-AUC;

2. Upload our codes.

These valuable suggestions will be dealt with before the deadline of the revision phase.

Next, we provide some detailed answers to some concrete questions. We are glad to see some further discussions to evaluate our work more comprehensively.

---

### Author Response · Authors · 2022-11-17
**General Response 2**

Dear Area Chairs and Reviewers,

We have made the following updates:

1) Upload the code in the supplementary materials.
2) Compare and discuss CBM-AUC, which is referred by reviewer k8KK, in the revised version.

We are glad to see your further replies and discussions.

---

### Author Response · Authors · 2022-11-18
**General Response 3**

Dear reviewers,

We have tried to address all your concerns of our paper and uploaded a revised version. Since there is only one day left for the discussion period to end, we would like to know if you have any other concerns in light of revision? We are ready to further update the pdf.


Sincerely,

Authors

---

### Author Response · Authors · 2022-12-08
**To AC: Further Questions after the Internal Discussion are Welcomed!**

Dear Area Chairs,

Considering the deadline of the current stage is approaching, we are looking forward to your internal discussion about our paper (if not done yet). It would be greatly appreciated if we could have the opportunity to reply to your further questions after the internal discussion.

Sincerely,

Paper 333 Authors

---

### Decision · Program_Chairs · 2023-01-20

**Decision:**

Reject

**Justification For Why Not Higher Score:**

The authors did not investigate the leakage between implicit concepts and explicit concepts, which might make the intervention result less reliable. Also, the idea of having implicit concepts to achieve higher accuracy is not novel, and the comparisons are not entirely fair (CBM-AUC; SAWADA et al.).

**Justification For Why Not Lower Score:**

The motivation is good, and the experiments results are good as many baselines are compared.

**Metareview: Summary, Strengths And Weaknesses:**

(a) The paper examines one issue for concept bottleneck models where there is insufficient concept information. They argue that in these cases, the concept bottleneck model may either have less predictive power or have concept distortion (only when concept and model are trained jointly). They thus propose to design a CBM model that has implicit concepts to retain high predictive performance, and also algorithms based on mutual information to automatically correct labels and trace back to incorrect concepts. Based on this a human may intervene to correct incorrect concept definitions. (b) It tackles one major issue of concept bottleneck model, and also propose an intervention where human can interact with the explanation (c) One weakness of the paper is that the implicit concepts does not address the concept leakage issue. Since this work attacks CBM that the explicit concepts may include information of implicit concepts (which leads to concept distortions), it is natural to ask the question: does the proposed DCBM's explicit concept also include information of implicit concepts. This question is not addressed by the author, which raises questions on the significance of the proposed method. Another issue is that the idea of separating concepts into supervised concepts and unsupervised concepts has been proposed by [Concept Bottleneck Model With Additional Unsupervised Concepts, IEEE, 2022 April]. While DCBM shows better performance on label accuracy, the implicit concepts in CBM-AUC is more interpretable than the implicit concepts in DCBM. The contribution of DCBM seems to overlap with the contribution of CBM-AUC.


**Summary Of Ac-Reviewer Meeting:**

One major issue of the AC-reviewer discussion is that the reviewers which raised questions regarding the concept leakage did not join the discussion.
All: Concept leakage issue may devalue CBM research that did not address this leakage slightly, and this submission's main point is also central to concept leakage (-)
K8KK: I think the contribution to the present SOTA of the concept-based models is fair, good experiments (+)  the paper is a bit hard to follow (-) novelty in the theoretical results presented (+)
kLNW (AC): Unfortunately, since not enough reviewers joined the discussion, I have to make the decision based on some personal judgement. I do feel like the concept leakage issue from explicit and implicit concepts is a valid concern, since the key of the paper is to decouple implicit concepts from explicit concepts to get better accuracy. If there is leakage between implicit concepts and explicit concepts, I am not sure the intervention result for DCBM is as reliable as the concept bottleneck one trained with sequential strategy (which does not have concept distortions but has lower accuracy), since the implicit concepts might still include some explicit concept information and make the intervention result a bit unreliable. I believe the manuscript will greatly benefit from a closer investigation into the leakage between implicit concepts and explicit concepts. While the concept leakage problem is a relative open question to most concept learning works(and concept bottleneck models), it is especially crucial to the decoupling concept direction.